# Rootstock effects on floral induction in commercial Iranian almond cultivars: Insights from morphophysiological, biochemical, and molecular analyses

Masoud Abedian-Chermahini[1], Behrouz Shiran[1,2]*, Abdolrahman Mohammadkhani[3], Habibollah Nourbakhsh[4]

1 Department of Plant Breeding and Biotechnology, Faculty of Agriculture, Shahrekord University, Shahrekord, Iran, 2 Institute of Biotechnology, Shahrekord University, Shahrekord, Iran, 3 Department of Horticulture, Faculty of Agriculture, Shahrekored University, Shahrekord, Iran, 4 Baghban-e-Zayandeh Rood Co., Saman, Chahrmahal va Bakhtiari, Iran

* shiran@sku.ac.ir, beshiran45@gmail.com

## Abstract

Orchard productivity in almond trees is strongly influenced by rootstock selection, which plays a key role in floral induction and yield optimization. Although several rootstocks are commonly used in Iran, their comparative effects on floral induction in major commercial cultivars remain poorly understood. This study evaluated five Peach×Almond hybrid rootstocks (GN15, GF677, GN15-M, Shurab2, and Shurab3) grafted with two widely grown Iranian cultivars, Mamaee and Shahrud12, to investigate rootstock–scion interactions. These combinations were chosen based on their commercial importance and regional adaptability. A four-year factorial experiment (2021–2025) was conducted in a completely randomized design. Morphological traits, including flower number, blooming density, and vegetative growth, were measured alongside photosynthetic pigments and endogenous hormone profiles. Additionally, the expression of flowering-related genes was analyzed in leaf and bud tissues. Results revealed that Shurab3 significantly enhanced floral induction in both cultivars, with the Shurab3–Shahrud12 combination producing the highest flower number and bloom density. Shurab3 also outperformed GN15 in promoting flowering in Mamaee, whereas the GN15–Mamaee combination showed the lowest performance. Rootstocks GF677, GN15-M, and Shurab2 exhibited intermediate effects. Shurab3 combinations were further associated with higher chlorophyll content, increased indole-3-acetic acid (IAA), and dynamic patterns of abscisic acid (ABA) and gibberellic acid (GA3). Molecular analyses confirmed upregulation of *FLOWERING LOCUS T* (*FT*), *CONSTANS* (*CO*), *SUPPRESSOR OF OVEREXPRESSION OF CO 1* (*SOC1*), *LEAFY* (*LFY*), and *APETALA1* (*AP1*) in Shurab3–Shahrud12, consistent with observed phenotypic improvements. Overall, these findings indicate that both rootstock and scion selection critically influence reproductive performance. Shurab3

**Data availability statement:** All relevant data are within the manuscript and its Supporting Information files.

**Funding:** The author(s) received no specific funding for this work.

**Competing interests:** The authors have declared that no competing interests exist.

emerges as a promising flower-inducing rootstock, providing practical guidance for optimizing orchard management and enhancing almond productivity under regional climatic conditions.

## Introduction

Rootstocks are fundamental in shaping the physiological and reproductive performance of fruit trees, affecting characteristics such as vegetative growth, yield potential, flowering patterns, and resilience to environmental stresses [1–5]. Among these traits, flowering is particularly critical, as it directly influences fruit set and overall productivity. Consequently, selecting an appropriate rootstock is a key aspect of orchard management, not only for optimizing tree vigor and adaptability but also for regulating the shift from vegetative to reproductive growth. Research on various *Prunus* species has shown that rootstocks can significantly modulate the expression of flowering-related genes, thereby impacting floral initiation and fruit yield [3,6–10]. For example, grafting onto specific rootstocks has been reported to enhance early flower bud differentiation and induce precocious flowering [11–13]. For almond (*Prunus dulcis*), the timing and extent of flower-bud differentiation are primary determinants of yield; therefore, rootstock selection is not only a tool to manage vigor and soil adaptation, but also a potential lever to regulate the shift from vegetative to reproductive development [14,15]. Recent transcriptomic and hormone-profiling studies in *Prunus* species indicate that rootstock genotype can alter scion gene expression programs related to architecture and developmental timing, suggesting a molecular basis for rootstock-mediated differences in flowering behaviour [3,14]. Comparable effects have been observed in other fruit crops, including apple, mango, and avocado, where rootstock–scion interactions influence flowering time, shoot maturation, and fruit set [16]. Despite these insights, the physiological and molecular mechanisms underlying rootstock-mediated regulation of flowering in almond remain poorly understood.

Almond (*Prunus dulcis* Mill.) is among the most economically significant nut crops worldwide, with global production surpassing 1.5 million metric tons in 2023–2024, predominantly contributed by the United States, the European Union, Australia, and Iran. To meet increasing global demand, continuous enhancement of yield efficiency is essential, particularly through the optimization of rootstock–scion combinations and the regulation of flowering processes. Rootstocks can influence flowering through multiple physiological routes, by modifying carbohydrate partitioning, hydraulic conductance, nutrient supply, and long-distance hormonal signals (e.g., ABA, IAA, gibberellins) that are known to affect floral induction and bud differentiation [14,15]. Consequently, understanding how rootstocks influence floral induction and development is therefore crucial for improving productivity and ensuring the long-term sustainability of almond orchards.

Flower induction and blossom formation in almond are intricate processes governed by the interaction of genetic, hormonal, and environmental factors [17–21]. Although almonds are typically classified as day-neutral species, floral differentiation is strongly influenced by internal cues such as carbohydrate status, hormonal

balance (e.g., ABA, IAA, and GA$_3$), and overall tree vigor [22–23]. These endogenous signals are further modulated by the rootstock, which can affect hormonal transport and metabolic communication between the root system and the scion canopy.

At the molecular level, advances in plant genomics have revealed several conserved flowering-related genes in *Arabidopsis*, whose homologs perform similar regulatory functions in perennial fruit trees [24–26]. Among these, the *FT* gene and its associated regulators (*CO*, *FD*, *SOC1*, *AP1*, and *LFY*) form the central framework of the floral induction network [27–29]. Nevertheless, the degree to which these molecular pathways are influenced by different almond rootstocks, particularly across the diverse genetic backgrounds of commercial cultivars, remains largely uncharacterized. The recent availability of improved almond genome assemblies and transcriptomic datasets now enables more precise mapping of gene-level responses to rootstock genotype [30,31].

In Iran, commercial almond production commonly employs peach–almond hybrid vegetative rootstocks (including GN15 and GF677) to enhance adaptability and compatibility. Locally developed rootstocks and clonal variants (e.g., GN15-M, Shurab2, Shurab3) are used in regional practice, yet published information on how these specific rootstocks influence floral induction and bud development in Iranian cultivars is limited. Addressing this gap will help producers choose rootstocks that improve floral initiation and yield stability under local growing conditions. Therefore, this study aimed to evaluate the effects of five rootstocks, GN15, GF677, GN15-M, Shurab2, and Shurab3, on floral induction in two Iranian commercial almond cultivars, Mamaee and Shahrud12. Specifically, the study sought to identify the most effective scion–rootstock combinations for enhancing flower induction and bud development.

The rootstocks used in this study included five peach × almond hybrids with different genetic backgrounds and geographical origins, namely GN15, GF677, GN15-M, Shurab2, and Shurab3. GN15 and GF677 are well-known interspecific hybrids of *Prunus persica × Prunus amygdalus* that were originally developed in Spain and France, respectively, and are widely used as vegetative rootstocks for almond production due to their vigor, adaptability to calcareous soils, and tolerance to drought and chlorosis [32,33]. GN15-M represents an asexually propagated mutant derived from GN15, selected in Spain for improved uniformity and vegetative propagation efficiency [33]. The Shurab2 and Shurab3 rootstocks, reported here for the first time, are newly developed Iranian peach × almond hybrids selected by the Baghban-e-ZayandehRood Company for their high adaptability to arid conditions and compatibility with local almond cultivars. The scion cultivars used were 'Mamaee' and 'Shahrud12', two commercial Iranian almond cultivars widely cultivated for their high yield, kernel quality, and adaptation to regional climatic conditions (Table 1) [34].

Beyond their relevance to Iranian almond production, the insights gained from this research may also contribute to broader rootstock selection programs in other semi-arid and Mediterranean-type regions. By elucidating how specific physiological and molecular traits of rootstocks influence floral induction, the findings can inform global breeding efforts aimed at developing rootstocks that optimize flowering performance and yield stability under diverse environmental conditions.

**Table 1. Pomological description, pedigree and origin of the rootstocks and scion cultivars used in the study.**

| Type | Genotype/ Cultivar | Pedigree | Origin | Pomological Description |
|------|--------------------|----------|--------|--------------------------|
| Rootstock | GN15 | Peach × Almond hybrid (vegetative rootstock) | Spain | Vegetative, used as rootstock for almonds |
| Rootstock | GF677 | Peach × Almond hybrid (vegetative rootstock) | France | Vegetative, used as rootstock for almonds |
| Rootstock | GN15-M | an asexually propagated mutant of GN15 | Spain | Vegetative, used as rootstock for almonds |
| Rootstock | Shurab2 | Peach × Almond hybrid (vegetative rootstock) | Iran | Vegetative, used as rootstock for almonds |
| Rootstock | Shurab3 | Peach × Almond hybrid (vegetative rootstock) | Iran | Vegetative, used as rootstock for almonds |
| Scion | Mamaee | Iranian commercial almond cultivar | Iran | Commercial cultivar, widely cultivated |
| Scion | Shahrud12 | Iranian commercial almond cultivar | Iran | Commercial cultivar, widely cultivated |

In summary, this study aims to elucidate how rootstock genotype affects floral induction and bud development in Iranian almond cultivars by integrating morphological, physiological, hormonal, and molecular analyses, providing insights into the mechanisms of rootstock–scion interactions and their implications for improving flowering efficiency and orchard productivity.

## Materials and methods

### Study site and experimental conditions

This study was conducted at the research orchard of Baghban-e-ZayandehRood Company, located within the Chaharmahal and Bakhtiari Research and Technology Park in Shurab-e-Saghir village, Saman County, Chaharmahal and Bakhtiari Province, Iran (32°30′38.99″N, 50°56′14.75″E; 2016 m above sea level). The region is characterized by cold winters, with minimum temperatures reaching approximately –10 °C, and warm summers, with maximum temperatures around 30 °C. The mean annual temperature ranges from 10 °C to 15 °C, and the average annual precipitation is 300–500 mm.

### Experimental design and plant material

A four-year experiment (April 2021–April 2025) was conducted to evaluate the effects of rootstock on flower induction in two Iranian commercial almond cultivars (Mamaee and Shahrud12). Each cultivar was grafted onto five rootstocks: GN15, GF677, GN15-M, Shurab2, and Shurab3 (Table 1). Rootstocks were planted in April 2021 at 2 m × 2 m spacing in sandy-loam soil and irrigated using a drip irrigation system. The experiment followed a factorial arrangement (rootstock × scion) in a completely randomized design (CRD) with six replications (Fig. 1).

### Grafting procedure

Rootstock stems approximately 1 cm in diameter were selected, and scions were obtained from vigorous, healthy plants with buds free of pests and disease. Grafting was performed in September 2021 using the T-budding technique. A T-shaped incision was made on the rootstock by intersecting horizontal and vertical cuts, and the scion bud was carefully inserted beneath the bark at the incision site. To prepare the scion, two angled cuts (~45°) were made above and below the bud before excision from the donor branch.

### Sampling and measurements

Morphological, physiological, biochemical, and molecular assessments began in the third year (2024). Because floral induction in almond typically begins in June, leaf samples were collected at four stages (June, July, August, and September 2024), and floral bud samples were collected at five stages (June, July, August, September, and December 2024) for laboratory analyses. Photosynthetic pigments were quantified using leaf samples collected in June, representing the midpoint of the growing season.

To analyze differential expression of flowering-related genes, specific sampling stages were selected: leaf samples from the second (July) and fourth (September) stages, and floral bud samples from the second (July), fourth (September), and fifth (December, dormancy) stages were used for gene expression analysis (Fig. 2). All samples were collected on the 10th of each month between 06:00 and 10:00 a.m., immediately frozen in liquid nitrogen, and stored at –80 °C until analysis.

**Morphological evaluation.** In the third year (2024), three comparable scaffolds were selected from each tree to evaluate flower induction and blooming. The number of flowers on each scaffold was recorded, and blooming density was calculated as the number of flowers per meter of scaffold length [35]. (A scaffold was defined as a major structural branch arising from the trunk that forms the primary framework of the tree).

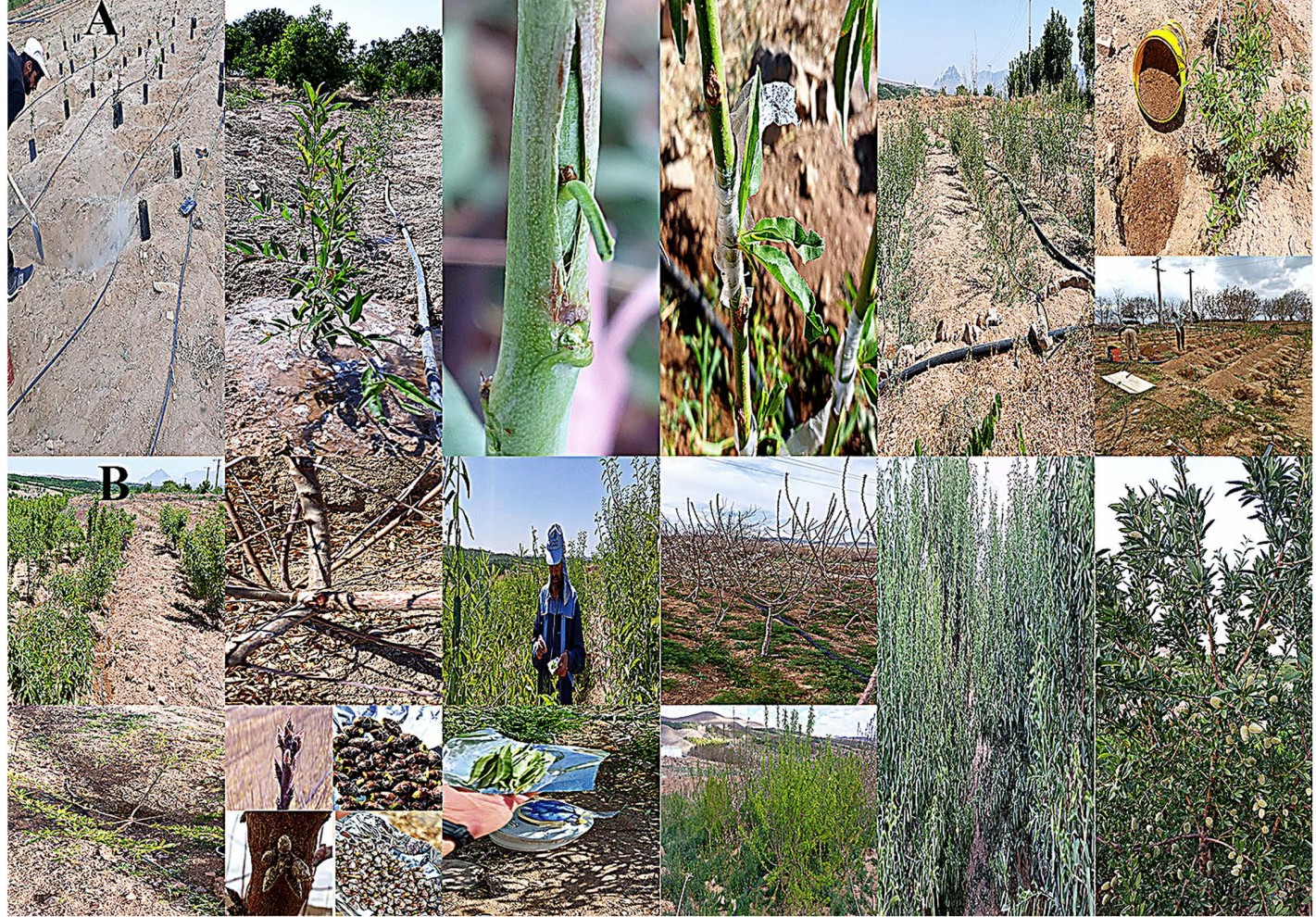

**Fig 1. Stages of the research process, from top left to bottom right (A) Planting, grafting, maintenance (B) Sampling, and trait measurement.**

Tree height and scaffold length were measured, and each scaffold was assessed for the number of nodes, internodes, and lateral branches. The mean internode length (mm) was then calculated. Trunk diameter was measured 10 cm above the graft union using a digital caliper, and scaffold diameters were recorded at their bases [36]. Trunk circumference was calculated using the formula:

$$\text{Circumference} = 2\pi r = 2\pi \times \frac{\text{diameter}}{2}$$

The trunk cross-sectional area (TCSA) and scaffold cross-sectional area (SCSA) were determined according to:

$$\text{TCSA or SCSA} = \pi r^2 = \pi \times \left(\frac{\text{diameter}}{2}\right)^2$$

All measurements were repeated in the fourth year (2025).

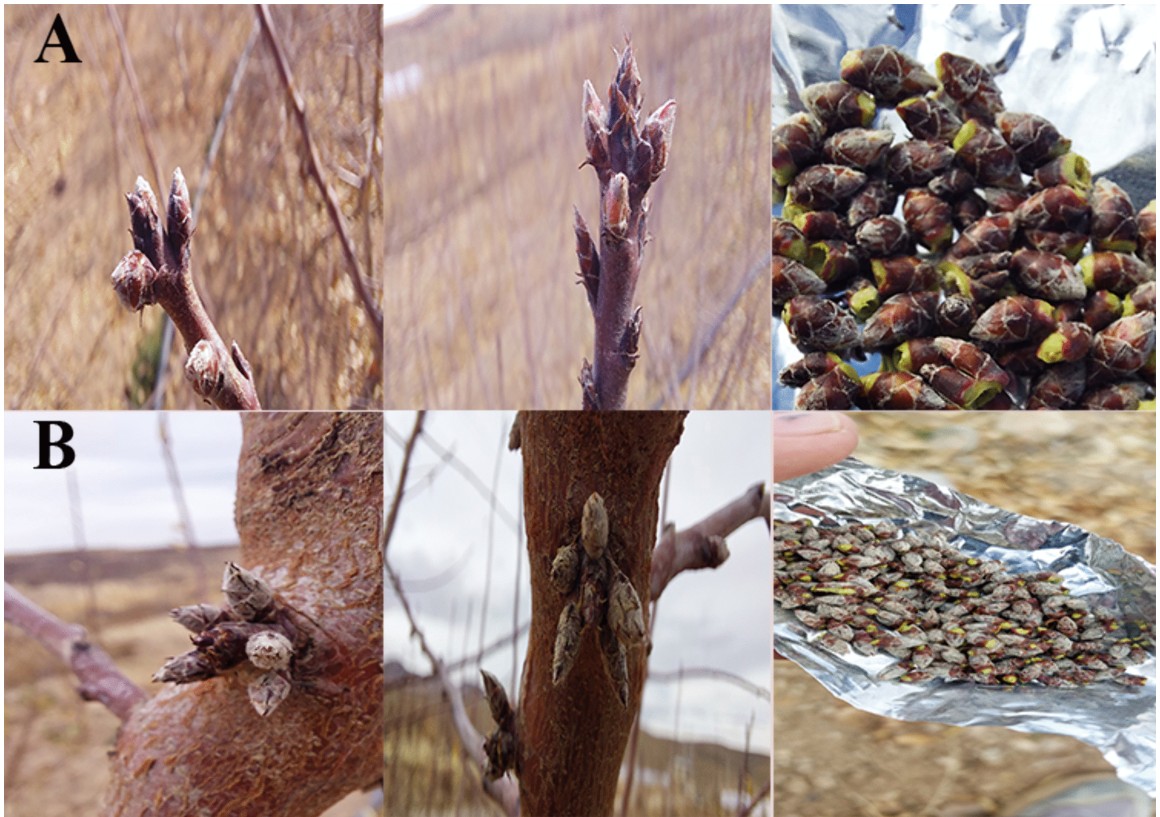

**Fig 2. Dormant flower buds of almond cultivars: (A) Mamaee almond cultivar; (B) Shahrud12 almond cultivar.**

**Photosynthetic pigment measurement.** Photosynthetic pigments were quantified from 0.1 g of frozen leaf tissue collected at sampling stage 1. The samples were homogenized in 10 mL of 80% (v/v) acetone and centrifuged at 1700 × g for 5 min. The resulting supernatant was used to determine chlorophyll *a* (Chl *a*), chlorophyll *b* (Chl *b*), total chlorophyll, and carotenoid contents. To prevent light-induced pigment degradation, all containers were wrapped in aluminum foil.

The absorbance of the extracts was measured at 645, 663, and 470 nm using an AE-S80-TS UV/Vis spectrophotometer (AELAB Guangzhou Co., Ltd). Pigment concentrations were calculated according to the equations of Arnon (1949) for chlorophylls and Lichtenthaler (1987) for carotenoids [37,38]:

$$\text{Chl a (mg g}^{-1}\text{ FW)} = [(12.7 \times A_{663}) - (2.69 \times A_{645})] \times \frac{V}{1000W}$$

$$\text{Chl b (mg g}^{-1}\text{ FW)} = [(22.9 \times A_{645}) - (4.68 \times A_{663})] \times \frac{V}{1000W}$$

$$\text{Total Chl (mg g}^{-1}\text{ FW)} = [(8.02 \times A_{663}) + (20.2 \times A_{645})] \times \frac{V}{1000W}$$

$$\text{Carotenoids (mg g}^{-1}\text{ FW)} = [\frac{(1000 \times A_{470}) - (1.82 \times \text{Chl a}) - (85.02 \times \text{Chl b})}{198}] \times \frac{V}{1000W}$$

where: $V$ = final extract volume (mL, in 80% acetone); $A$ = optical absorbance at the specified wavelength; $W$ = fresh tissue weight (g).

## Endogenous hormone measurement

Based on morphological evaluations, rootstock–scion combinations showing the highest and lowest levels of flowering induction and flower bud abundance were selected for quantification of endogenous hormone levels in floral buds across five sampling stages.

The concentrations of abscisic acid (ABA), indole-3-acetic acid (IAA), and gibberellic acid ($GA_3$) in floral buds were determined according to the protocol described by Arteca (1995) [39], using three biological replicates per sample. Approximately 1 g of floral bud tissue was homogenized in 10 mL of 80% ethanol in a prechilled mortar. The homogenate was shaken for 6 h and incubated overnight at 5 °C. The mixture was filtered through filter paper and centrifuged at 17,877 × $g$ for 10 min at 4 °C. The supernatant was concentrated, and its pH was adjusted to 8.8 with 2.5 N sodium hydroxide. The aqueous phase was partitioned with ethyl acetate in a separatory funnel. The solution pH was then adjusted to 2.4 using 0.4 N hydrochloric acid, followed by a second extraction with ethyl acetate. The ethyl acetate layer, containing the target hormones, was collected, and the solvent was evaporated using a rotary evaporator. The resulting dry residue was redissolved in 2 mL of ethanol and stored at –20 °C until analysis.

Hormone quantification was performed using high-performance liquid chromatography (HPLC; Agilent 1090, Agilent Technologies, USA) equipped with a diode array detector (DAD). Separation was achieved on a Waters Symmetry C18 column (150 × 4.6 mm, 5 μm) using ethanol as the mobile phase at 20 °C. Detection wavelengths were set at 280 nm for IAA, 206 nm for $GA_3$, and 265 nm for ABA. Analytical-grade standards were obtained from Sigma-Aldrich (Canada). Calibration curves (0.1–100 ng mL$^{-1}$) were used to assess linearity and detection limits. All analyses were conducted under controlled laboratory conditions (22 °C) to minimize variability.

## RNA extraction and RT-quantitative PCR analysis

Gene expression associated with flowering induction was analyzed using leaf samples and floral bud samples. Based on morphological evaluations, rootstock–scion combinations exhibiting the highest and lowest levels of flowering induction and flower bud abundance were selected for analysis. Ungrafted rootstocks were included as controls.

Total RNA was extracted from 100 mg of frozen tissue following the protocol of Rubio-Piña and Zapata-Pérez (2011) [40,41]. Briefly, samples were ground to a fine powder in liquid nitrogen, and 1.1 mL of extraction buffer (0.1 M Tris-HCl, pH 8.0; 20 mM EDTA, pH 8.0; 1.4 M NaCl; 2% w/v CTAB; 2% w/v PVP; 80 μL β-mercaptoethanol) was added in 2 mL microcentrifuge tubes. The mixtures were incubated at 65 °C for 10 min, followed by the addition of 800 μL chloroform, brief vortexing, and centrifugation at 11,180 × $g$ for 10 min at 4 °C. The supernatant was sequentially extracted with equal volumes of phenol/chloroform (1:1) and chloroform/isoamyl alcohol (24:1). RNA was precipitated with 8 M LiCl at −20 °C for 4 h and centrifuged at 13,528 × $g$ for 20 min at 4 °C. The resulting RNA pellets were washed, air-dried, and dissolved in DEPC-treated water.

RNA concentration and purity were assessed using a Biophotometer (Eppendorf, Germany), and integrity was confirmed by non-denaturing agarose gel electrophoresis. Genomic DNA contamination was removed by DNase I treatment. First-strand cDNA was synthesized using the YTA Reverse Transcriptase Kit (Yekta Tajhiz Azma, Iran), following the manufacturer's protocol.

Five flowering-related genes and one reference gene were selected for RT-quantitative PCR (RT-qPCR) analysis. The target genes included *FLOWERING LOCUS T (FT)*, *SUPPRESSOR OF OVEREXPRESSION OF CO1 (SOC1)*, and *CONSTANS (CO)*, primarily expressed in leaves, and *LEAFY (LFY)* and *APETALA1 (AP1)*, predominantly expressed in floral buds [42,43]. *Actin* (*PdActin*) was used as the internal reference gene. Primers were designed using Primer3 software and validated for specificity with NCBI Primer–BLAST [44]. The sequences of primers used in the RT-qPCR assay are listed in Table 2.

**Table 2. Primer sequences used for RT-qPCR in this study.**

| Gene Name | Gene ID | Forward Primer (5'- > 3') | Reverse Primer (5'- > 3') | Product Size(bp) |
|---|---|---|---|---|
| CO | LOC117628416 | GTGCGATCCTTACACCAAGT | GCTTCTCTCTGTACCTCAACAC | 110 |
| AP1 | LOC117624177 | GAGAAAGGAAAGGGCGATGC | ACATGGAAGAGGCTGTGGAA | 142 |
| FT | LOC117631186 | GGCAATTGGGTAGGCAAACA | CTTCTCCTTCCTCCAGAGCC | 148 |
| LFY | LOC117627698 | TTCTCAGCGAGCCTCTTCAA | TGCCGTGTAGTATCTGACCC | 189 |
| SOC1 | LOC117617955 | CAGATGAGGCGCATAGAGAAC | AACCTCAGCATCACAGAGAAC | 111 |
| PdAct | LOC117629987 | AGCGGGAAATTGTCCGTGAT | AAGAGAACTTCTGGGCACCG | 172 |

CO: CONTANS, AP1: APETALA1, FT: FLOWERING LOCUS T, LFY: LEAFY, SOC1: SUPPRESSOR OF OVEREXPRESSION OF CO 1, PdAct: Pd Actin

RT-qPCR reactions were performed using YTA SYBR Green PCR Master Mix (Yekta Tajhiz Azma, Iran) with two biological and three technical replicates per sample. Relative gene expression levels were normalized to *Actin* expression. Quantification of relative transcript abundance was performed using the $2^{-\Delta\Delta CT}$ method, as described by Livak and Schmittgen (2001) [45].

## Statistical analysis

Before performing statistical analyses, the assumptions underlying analysis of variance (ANOVA) were verified to ensure the validity of the results. Data normality was tested using the Shapiro–Wilk test, and homogeneity of variances was assessed with Levene's test. Both assumptions were satisfied, confirming the suitability of ANOVA for the dataset. All statistical tests were conducted at a significance level of $\alpha = 0.05$. Analyses were performed using SAS software version 9.4 (SAS Institute Inc., Cary, NC, USA). The procedures included ANOVA at the 5% significance level, mean comparisons using the least significant difference (LSD) test, and Spearman's rank correlation analysis. Principal component analysis (PCA) was conducted with STATGRAPHICS Centurion version 19 (Statgraphics Technologies, Inc., The Plains, VA, USA). Graphs were created using Microsoft Excel (Microsoft Corp., Redmond, WA, USA) and GraphPad Prism version 10.4 (GraphPad Software, San Diego, CA, USA).

## Results

### Morphological evaluation

**Analysis of variance for morphological traits.** Analysis of variance (ANOVA) was performed for morphological traits, including the number of flowers per scaffold, blooming density, number of nodes and internodes per scaffold, internode length, tree height, scaffold length, number of lateral branches per scaffold, trunk cross-sectional area (TCSA), scaffold cross-sectional area (SCSA), and trunk circumference, in two Iranian commercial almond cultivars (Mamaee and Shahrud12) grafted onto five rootstocks. The detailed ANOVA results are presented in Table 3.

Rootstock, scion, and their interaction had significant effects across both growing seasons (2024–2025). Rootstock exerted a highly significant influence ($P \leq 0.01$) on all evaluated traits in both years. In 2024, the largest mean squares were observed for the number of flowers per scaffold and blooming density, whereas trunk circumference exhibited the lowest significant effect. By 2025, the influence of rootstock became even more pronounced, particularly for flower number and blooming density, indicating a year-to-year intensification of rootstock effects. Vegetative traits such as tree height, scaffold length, and the number of lateral branches also increased significantly in response to rootstock type.

Scion effects varied depending on the trait and year. In 2024, the scion significantly influenced flower number, number of nodes, and tree height but had no significant effect on TCSA or SCSA. By 2025, the influence of scion became more pronounced, particularly for flower number and scaffold length, with additional significant effects detected for the number

**Table 3. ANOVA results for morphological traits including number of flowers per scaffold, blooming density, number of nodes, and internodes per scaffold and internode length of two Iranian almond cultivars grafted onto five rootstocks over two growing seasons (2024–2025) based on LSD test at P<0.05.**

Mean Square

| Source of variation | DF | No. Flowers per Scaffold | | Blooming Density | | No. Node per Scaffold | | No. Internode per Scaffold | | Internode Length | |
|---|---|---|---|---|---|---|---|---|---|---|---|
| | | 2024 | 2025 | 2024 | 2025 | 2024 | 2025 | 2024 | 2025 | 2024 | 2025 |
| Rootstock | 4 | 114627.9** | 1125091.18** | 32764.27** | 184699.17** | 172.548** | 351.97** | 172.55** | 351.97** | 20.02** | 18.55** |
| Scion | 1 | 256498.8** | 1303310.81** | 48319.78** | 65281.93** | 7874.85** | 20438.17** | 7874.85** | 20438.17** | 20.96** | 26.28** |
| Rootstock×scion | 4 | 29034.59** | 84629.66*** | 3599.88* | 20593.56*** | 1021.21** | 2588.35** | 37.15** | 2588.35** | 14.16** | 49.37** |
| Error | 50 | 4638.266 | 4598.23 | 1132.23 | 484.46 | 27.48 | 26.23 | 27.48 | 26.23 | 1.65 | 0.84 |
| CV% | | 33.89 | 10.19 | 27.23 | 6.66 | 5.31 | 3.74 | 5.37 | 3.77 | 8.03 | 6.36 |

**, * and ns; significant at 0.01, 0.05 probability levels and not significant, respectively, CV: Coefficient of variation

ANOVA results for morphological traits, including tree height, scaffold length, number of lateral branches per scaffold, TCSA, SCSA, and trunk circumference of two Iranian almond cultivars grafted onto five rootstocks over two growing seasons (2024–2025) based on LSD test at P < 0.05.

| Source of variation | DF | Tree Height | | Scaffold length | | No. lateral Branches per Scaffold | | TCSA | | SCSA | | Trunk Circumference | |
|---|---|---|---|---|---|---|---|---|---|---|---|---|---|
| | | 2024 | 2025 | 2024 | 2025 | 2024 | 2025 | 2024 | 2025 | 2024 | 2025 | 2024 | 2025 |
| Rootstock | 4 | 1442.18* | 2890.93** | 2746.14** | 5019.52** | 514.33** | 1026.85** | 97.75** | 335.27** | 1.5* | 10** | 19.29** | 33.91** |
| Scion | 1 | 42613.35** | 91963.35** | 99750.05** | 18362.05** | 98.04ns | 234.82* | 48.13* | 72.95** | 0.55ns | 0.32ns | 7.29* | 13.17** |
| Rootstock×scion | 4 | 1866.26* | 4172.16** | 3437.11** | 16696.22** | 512.09** | 1137.71** | 147.55** | 353.59** | 10.57** | 36.16** | 22.95** | 45.38** |
| Error | 50 | 524.03 | 528.36 | 174.33 | 163.54 | 50.55 | 48.63 | 7.21 | 6.84 | 0.52 | 0.48 | 1.28 | 1.18 |
| CV% | | 10.02 | 8.6 | 8.5 | 6.58 | 22.5 | 13.19 | 15.42 | 9.16 | 17.2 | 10.91 | 7.74 | 5.99 |

**, * and ns; significant at 0.01, 0.05 probability levels and not significant, respectively. TCSA: Trunk cross-sectional area, SCSA: scaffold cross-sectional area, CV: Coefficient of variation

of lateral branches and trunk circumference. Blooming density was significantly affected by scion in 2024; however, this effect was moderated in 2025.

The rootstock × scion interaction was significant for most traits (P ≤ 0.01 or P ≤ 0.05), except for the number of lateral branches in 2024, which was not significant (98.04 ns) (Table 3).

**Mean values of morphological and floral traits.** Mean values of morphological and floral traits for the ten rootstock–scion combinations over two consecutive years (2024 and 2025) are presented in Table 4 (Supporting information). All traits differed significantly among genotypes and between years (P < 0.05, LSD test) based on six replicates.

**Comparison of mean flowering traits among rootstock–scion combinations:** The highest and lowest flowering induction rates and number of flowers per scaffold were consistently observed in Shurab3–Shahrud12 (significantly greater than all other combinations) and GN15–Mamaee, respectively. Blooming density followed a similar pattern, with Shurab3–Shahrud12 exhibiting the highest values in both years and GN15–Mamaee the lowest. In Shurab3–Shahrud12, blooming density exceeded that of GN15–Mamaee by more than threefold (P < 0.05). Intermediate flower production was observed in combinations such as GF677–Shahrud12 and GN15M–Shahrud12. Among rootstocks, Shurab3 consistently promoted the greatest floral induction and flower number, followed by GF677, GN15-*M*, and Shurab2, whereas GN15 showed the lowest effectiveness.

**Comparison of mean vegetative traits among rootstock–scion combinations:** In terms of growth, Shurab3–Shahrud12 and Shurab2–Shahrud12 produced the tallest trees, while Shurab2–Mamaee exhibited the shortest stature. Scaffold length followed a similar trend. The number of nodes and internodes per scaffold was highest in Shurab3–Shahrud12 and lowest in Shurab2–Mamaee. Internode length varied significantly among combinations, with GN15M–Mamaee producing the longest internodes in both years. The greatest number of lateral branches per scaffold was observed in Shurab3–Shahrud12, significantly exceeding Shurab2–Mamaee.

Trunk cross-sectional area (TCSA) was highest in GN15M–Shahrud12 and lowest in Shurab2–Mamaee. Scaffold cross-sectional area (SCSA) followed a similar pattern, with Shurab2–Shahrud12 and Shurab2–Mamaee showing the highest and lowest values, respectively. Trunk circumference was largest in GN15M–Shahrud12 and smallest in Shurab2–Mamaee (Table 4).

## Photosynthetic pigment measurement

The concentrations of photosynthetic pigments (chlorophyll *a*, chlorophyll *b*, total chlorophyll, and carotenoids) across the ten rootstock–scion combinations are shown in Fig 3. Chlorophyll *a* content was highest in Shurab3–Shahrud12 (0.53 ± 0.01 mg g$^{-1}$ FW), significantly exceeding the lowest value observed in GN15M–Mamaee (0.33 ± 0.01 mg g$^{-1}$ FW). Chlorophyll *b* exhibited a similar pattern, with the highest concentration in GN15–Mamaee (0.88 ± 0.03 mg g$^{-1}$ FW) and the lowest in GN15M–Mamaee (0.24 ± 0.01 mg g$^{-1}$ FW). Total chlorophyll content followed the same trend, being maximal in GN15–Mamaee (0.83 ± 0.02 mg g$^{-1}$ FW) and minimal in GN15M–Mamaee (0.29 ± 0.01 mg g$^{-1}$ FW), with significant differences among combinations. Carotenoid concentrations ranged from 0.12 ± 0.01 mg g$^{-1}$ FW in GN15M–Mamaee to 0.18 ± 0.01 mg g$^{-1}$ FW in Shurab3–Shahrud12, Shurab3–Mamaee, GN15–Shahrud12, and GF677–Shahrud12, indicating considerable variation across genotypes (Fig. 3, Supporting information).

## Principal component analysis and clustering

Principal component analysis (PCA) was conducted using the morphological traits and photosynthetic pigment data from all rootstock–scion combinations during the 2024 and 2025 growing seasons. Four principal components (PCs) were extracted based on the Kaiser criterion (eigenvalues ≥ 1.0), collectively explaining 89.64% of the total variance in the dataset [46]. The first principal component (PC1) accounted for 40.01% of the variance (eigenvalue = 6.00) and was strongly associated with structural and productivity-related traits. Trunk cross-sectional area (TCSA), tree height, and scaffold length exhibited the highest positive loadings on this axis, indicating their major contribution to overall variation.

**Table 4. Mean±SE of morphological traits including, number of flowers per scaffold, and blooming density, number of node and internode per scaffold, internode length, with different lower-case letters indicating significance for ten rootstock-scion combinations measured in 2024 and 2025.**

| Rootstock×Scion | No. Flowers per Scaffold | | Blooming Density | | No. Node per Scaffold | | No. Internode per Scaffold | | Internode Length | |
|---|---|---|---|---|---|---|---|---|---|---|
| | 2024 | 2025 | 2024 | 2025 | 2024 | 2025 | 2024 | 2025 | 2024 | 2025 |
| GF677-Mamaee | 178.67± 15.04 cd | 574.67± 13.3ed | 122.38± 6.73ced | 319.66± 4.65c | 90.33± 0.89c | 126.33± 0.75e | 89.33± 0.89c | 125.33± 0.75e | 16.3± 0.59cb | 14.36± 0.36c |
| GF677-Shahrud12 | 262.78± 18.76b | 815.78± 17.02cb | 161.8± 11.53b | 389.9± 7.03b | 100.45± 1.59b | 143.45± 1.3c | 99.45± 1.59b | 142.45± 1.3c | 16.27± 0.24cbd | 14.7± 0.12cb |
| GN15-Mamaee | 64.67± 4.11f | 307.67± 2.37h | 43.89± 3.25g | 159.89± 2.76f | 95.39± 0.98cb | 135.39± 0.64d | 94.39± 0.98cb | 134.39± 0.64d | 15.78± 0.36cbd | 14.34± 0.21c |
| GN15-Shahrud12 | 133.28± 7.64fed | 425.28± 5.9gf | 99.36± 2.32fed | 253.67± 5.6d | 93.72± 3.03c | 132.72± 2.41d | 92.72± 3.03c | 131.72± 2.41d | 14.92± 0.31ced | 12.77± 0.22d |
| GN15M-Mamaee | 153.95± 7.59ed | 496.95± 5.85ef | 89.24± 8.32fe | 200.91± 8.4e | 89.89± 2.5c | 124.89± 1.9e | 88.89± 2.5c | 123.89± 1.9e | 20.26± 1.25a | 20.2± 0.9a |
| GN15M-Shahrud12 | 242.06± 21.92cb | 774.06± 20.18c | 134.48± 10.49cbd | 388.66± 4.3b | 116.06± 0.84a | 167.06± 0.65b | 115.06± 0.84a | 166.06± 0.65b | 15.44± 0.31cbd | 11.98± 0.18d |
| Shurab2-Mamaee | 80.33± 11.23fe | 346.33± 9.49gh | 74.34± 9.52fg | 269.79± 4.95d | 77.89± 1.96d | 107.89± 1.1f | 76.89± 1.96d | 106.89± 1.1f | 13.93± 0.61e | 12.04± 0.38d |
| Shurab2-Shahrud12 | 188.39± 17.52cbd | 611.39± 15.78d | 101.94± 9.22fed | 257.57± 8.1d | 118.5± 0.98a | 171.5± 0.61ba | 117.5± 0.98a | 170.5± 0.61ba | 14.79± 0.19ed | 13.95± 0.09c |
| Shurab3-Mamaee | 200.06± 17.99cbd | 878.06± 16.25b | 146.1± 11.86cb | 535.17± 8.4a | 82.33± 1.31d | 112.33± 0.94f | 81.33± 1.31d | 111.33± 0.94f | 16.77± 0.17b | 14.74± 0.08cb |
| Shurab3-Shahrud12 | 505± 76.04a | 1451± 74.3a | 262.14± 36.61a | 525.48± 15.6a | 121.67± 4.35a | 176.67± 2.65a | 120.67± 4.35a | 175.67± 2.65a | 15.71± 0.19cbd | 15.67± 0.09b |

All measurements were performed in 6 replicates. Different lower-case letters after mean values±SE indicate significant differences based on LSD test at P<0.05.

**Mean ± SE of morphological traits including, tree height, scaffold length, number of lateral branches per scaffold, TCSA, SCSA, trunk circumference, with different lower-case letters indicating significance for ten rootstock-scion combinations measured in 2024 and 2025.**

| Rootstock×Scion | Tree Height | | Scaffold length | | No. lateral Branches per Scaffold | | TCSA | | SCSA | | Trunk Circumference | |
|---|---|---|---|---|---|---|---|---|---|---|---|---|
| | 2024 | 2025 | 2024 | 2025 | 2024 | 2025 | 2024 | 2025 | 2024 | 2025 | 2024 | 2025 |
| GF677-Mamaee | 202.67± 7.92c | 232.67± 6.77e | 144.11± 6.14d | 180.11± 5.12fe | 35.22± 0.85cb | 60.22± 0.43cb | 16.05± 1.36e | 25.86± 1.12c | 4.56± 0.17b | 7.12± 0.16c | 14.13± 0.6c | 18.03± 0.5e |
| GF677-Shahrud12 | 234± 0.86b | 274± 0.75dc | 161.22± 1.08bc | 209.22± 0.98c | 28.67± 2.88 cd | 49.67± 1.56ed | 10.47± 0.18f | 17.78± 0.16d | 3.2± 0.25ed | 4.42± 0.22e | 11.47± 0.1d | 14.97± 0.16f |
| GN15-Mamaee | 191.17± 4.14c | 211.17± 3.96fe | 147.67± 2.2dc | 192.67± 1.95de | 35± 1.04cb | 58± 0.84cb | 18.66± 0.7ecd | 31.15± 0.72b | 4.05± 0.05cb | 6.1± 0.04d | 15.3± 0.3bc | 19.8± 0.32dc |
| GN15-Shahrud12 | 250.17± 3.48ba | 295.17± 3.12bc | 138.5± 7.1d | 168.5± 6.3fg | 19.44± 0.69e | 37.44± 0.43f | 16.29± 0.27ed | 27.32± 0.32c | 3.69± 0.08 cd | 5.38± 0.07d | 14.3± 0.12c | 18.5± 0.11de |
| GN15M-Mamaee | 230.5± 24.62b | 265.5± 23.89d | 177.78± 10.36a | 249.78± 9.56b | 37.78± 2.32b | 65.78± 1.23b | 17.66± 0.46ecd | 33.29± 0.51b | 4.8± 0.35b | 8.6± 0.34b | 14.89± 0.19bc | 19.29± 0.16dce |
| GN15M-Shahrud12 | 262± 4.46a | 317± 4.13ba | 177± 3.72a | 199± 2.97dc | 28.28± 0.57 cd | 48.28± 0.45ed | 23.27± 1.05a | 38.09± 1.06a | 4.26± 0.34cb | 7.06± 0.33c | 17.08± 0.39a | 22.08± 0.32a |
| Shurab2-Mamaee | 183.33± 2.25c | 198.33± 1.98f | 106.33± 2.44e | 128.33± 2.03h | 24.11± 0.76ed | 42.11± 0.62ef | 10.27± 0.18f | 17.34± 0.17d | 2.38± 0.08e | 3.48± 0.07f | 11.36± 0.1d | 14.76± 0.16f |
| Shurab2-Shahrud12 | 275.33± 1.93a | 335.33± 1.41a | 174.83± 2.9ba | 237.83± 2.4b | 24.5± 1.65ed | 43.5± 1.3ef | 22.27± 0.55ba | 37.09± 0.57a | 5.84± 0.47a | 9.64± 0.46a | 16.72± 0.21a | 21.52± 0.18ba |
| Shurab3-Mamaee | 201.67± 1.33c | 226.67± 1.03e | 136.06± 1.25d | 164.06± 0.95g | 32.22± 0.83cbd | 53.22± 0.74 cd | 19.94± 1.44bc | 33.37± 1.03b | 4.73± 0.08b | 7.46± 0.06c | 15.77± 0.59ba | 20.47± 0.46bc |
| Shurab3-Shahrud12 | 254.33± 11.99ba | 304.33± 10.2b | 189.33± 7.71a | 275.33± 6.01a | 50.67± 8a | 80.67± 6.98a | 19.23± 2.42bcd | 31.75± 1.23b | 4.5± 0.55cb | 7± 0.45c | 15.36± 1.04bc | 19.96± 0.86c |

All measurements were performed in 6 replicates. Different lower-case letters after mean values±SE indicate significant differences based on LSD test at P<0.05.

TCSA: Trunk cross-sectional area, SCSA: scaffold cross-sectional area

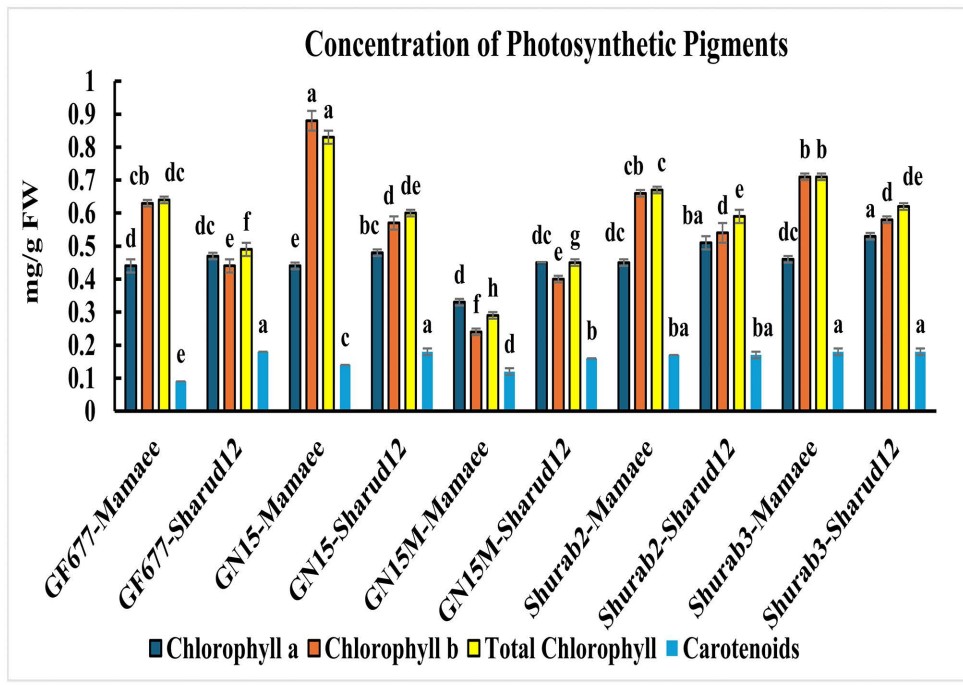

**Fig 3. The concentrations of photosynthetic pigments (chlorophyll a, chlorophyll b, total chlorophyll, and carotenoids) across ten rootstock-scion combinations. Columns with the same superscript letter are not significantly different (P < 0.05, LSD test).**

The second component (PC2) explained 23.15% of the variance (eigenvalue = 3.47) and was primarily associated with photosynthetic pigments. Chlorophyll *a*, chlorophyll *b*, total chlorophyll, and carotenoids showed strong positive loadings on PC2, reflecting their coordinated accumulation and contribution to photosynthetic capacity. The PCA biplot revealed clear differentiation among rootstock–scion combinations. Shurab3–Mamaee and GN15–Shahrud12 scored highly along PC2, indicating elevated pigment levels, whereas GN15M–Mamaee exhibited low PC2 scores, corresponding to reduced pigment accumulation.

The third principal component (PC3) accounted for 14.00% of the total variance (eigenvalue = 2.10) and was mainly associated with reproductive attributes, including the number of flowers and blooming density. The Shurab3–Shahrud12 combination exhibited a strong positive association along this axis, reflecting its superior flowering performance. The fourth component (PC4) explained 12.48% of the variance (eigenvalue = 1.87) and was primarily defined by internode length and the number of lateral branches per scaffold. Notably, GN15–Mamaee and GF677–Mamaee were distinctly separated along this component, indicating their characteristic vegetative growth patterns (Table 5, Fig. 4, S1 Table).

The biplot revealed distinct clustering patterns among the rootstock–scion combinations. Shurab3–Mamaee and Shurab2–Mamaee were positively associated with PC2, reflecting their elevated pigment contents, whereas GN15–Mamaee and GF677–Mamaee were positioned negatively along PC1, corresponding to lower structural trait values (Table 5; Fig. 4; S1 Table).

**Table 5. PCA of the morphological traits and photosynthetic pigments measured in ten rootstock-scion combinations during the 2024 and 2025 seasons.**

| Principal Components | PC1 | PC2 | PC3 | PC4 |
|---|---|---|---|---|
| Eigenvalues | 6.00168 | 3.47211 | 2.09996 | 1.87195 |
| Contribution ratio% | 40.011 | 23.147 | 14.000 | 12.480 |
| Cumulative contribution ratio % | 40.011 | 63.159 | 77.158 | 89.638 |

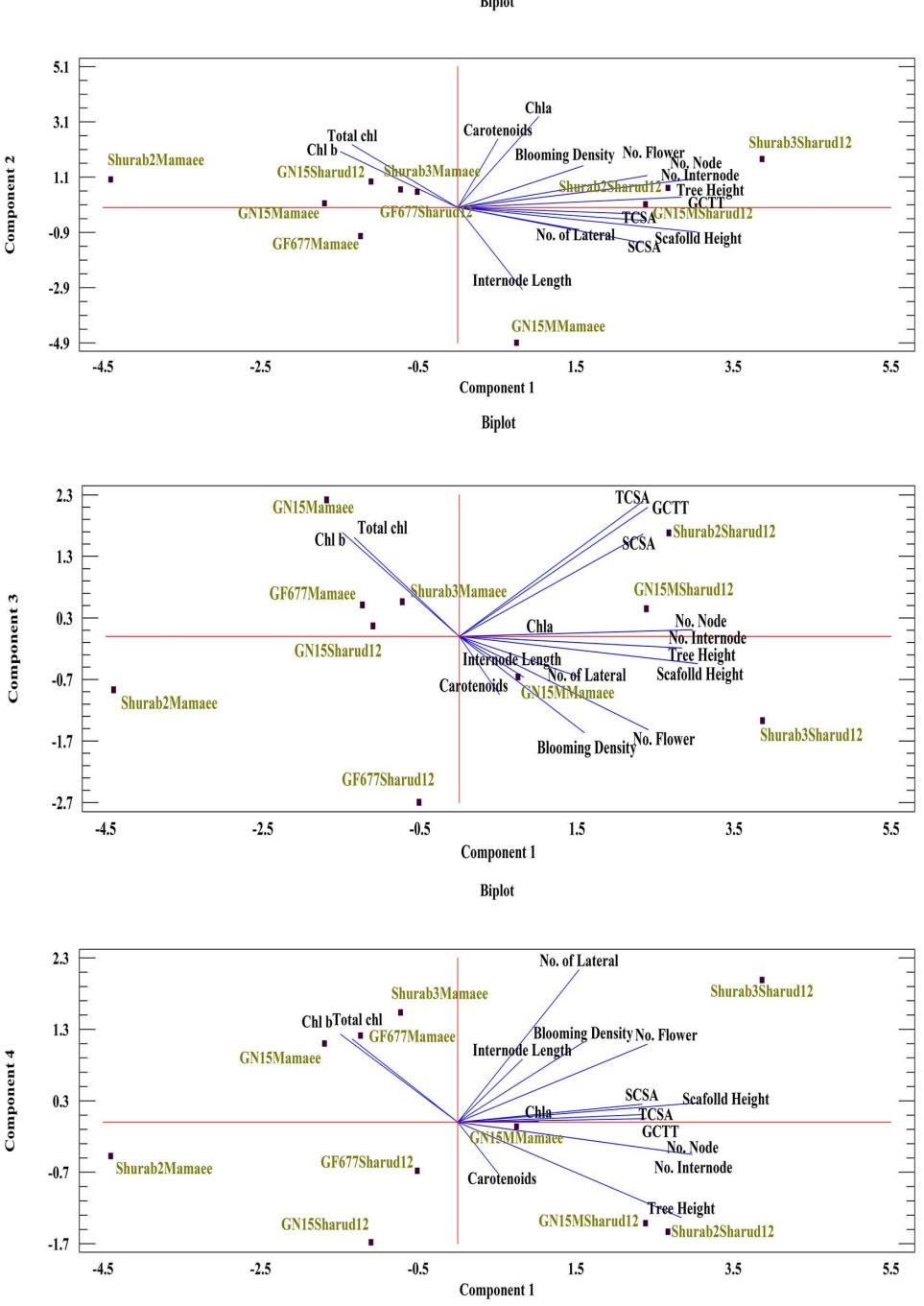

**Fig 4. Biplot of the morphological traits and photosynthetic pigments measured in ten rootstock-scion combinations during the 2024 and 2025 seasons.**

Collectively, the four principal components explained 89.64% of the total variance, providing a comprehensive representation of the dataset's multivariate structure. The remaining variance was likely attributable to environmental variation or other unmeasured factors.

Hierarchical clustering analysis, visualized via a heatmap, was conducted to assess the relationships among morphological traits and photosynthetic pigments (Fig. 5). The analysis grouped the rootstock–scion combinations based on their similarity and correlation patterns across the measured variables. The color gradient denotes the relative magnitude of each trait, with darker shades indicating higher values and lighter shades representing lower ones. The heatmap revealed distinct clustering trends: Shurab3–Shahrud12 and Shurab3–Mamaee formed a closely related cluster characterized by elevated values for the number of flowers per scaffold, blooming density, tree height, and scaffold length, as shown by the intense red shading. In contrast, GN15–Mamaee and Shurab2–Mamaee clustered together, exhibiting lower pigment concentrations and reduced morphological trait values, reflected by blue shading. The remaining combinations displayed intermediate trait levels, with GN15–Mamaee notably presenting relatively high chlorophyll *a* and *b* contents (Fig. 5).

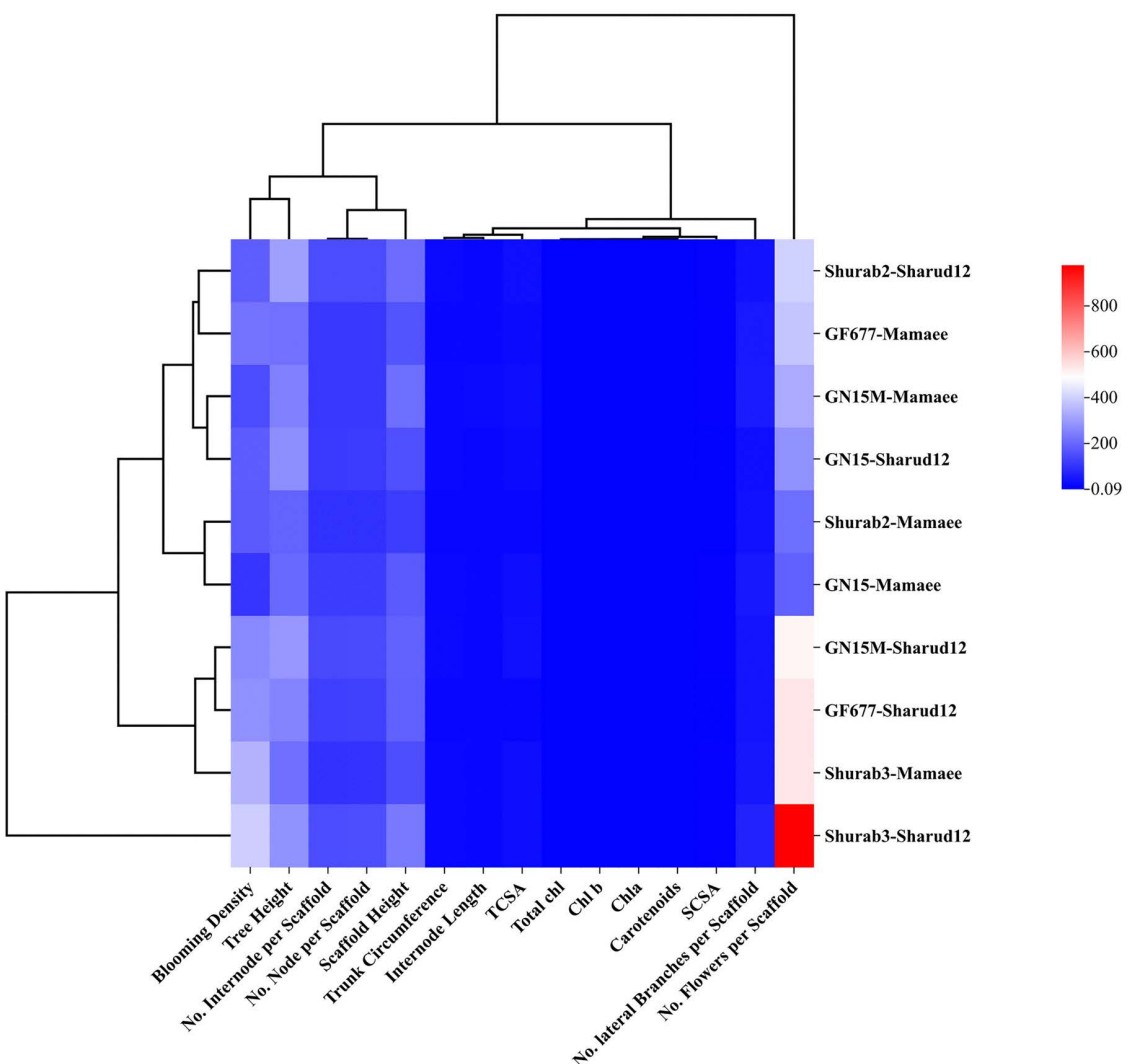

**Fig 5. Cluster heatmap of ten rootstock-scion combinations of commercial almond varieties that are grouped based on their similarity and their correlations with morphological traits and photosynthetic pigments.**

## Correlation matrix among morphological traits and photosynthetic pigments

The correlation matrix (Table 6) summarizes the relationships between morphological traits and photosynthetic pigments across the studied rootstock–scion combinations. Several significant positive and negative correlations were identified, providing insights into the interactions between structural growth and physiological performance. Tree height exhibited strong positive correlations with scaffold length (r = 0.76, $P < 0.01$), the number of nodes per scaffold (r = 0.71, $P < 0.01$), and the number of internodes per scaffold (r = 0.71, $P < 0.01$), indicating a close association between vertical growth and branching architecture. Likewise, the number of flowers per scaffold was highly and positively correlated with blooming density (r = 0.93, $P < 0.01$), underscoring the direct link between floral abundance and reproductive density. Moderate positive correlations were also observed between blooming density and both chlorophyll $a$ (r = 0.32, $P < 0.05$) and carotenoid content (r = 0.32, $P < 0.05$), suggesting a potential relationship between reproductive vigor and photosynthetic efficiency. Conversely, several other traits exhibited weak or nonsignificant correlations, implying that they may be influenced by independent physiological or genetic factors (Table 6).

## Endogenous hormones in floral buds

The quantification of endogenous hormone levels in floral buds from two contrasting rootstock–scion combinations, Shurab3–Shahrud12 (high flowering induction) and GN15–Mamaee (low flowering induction), revealed distinct patterns in gibberellic acid ($GA_3$), indole-3-acetic acid (IAA), and abscisic acid (ABA) concentrations across five developmental stages (stages 1–5) (Supporting information). In both combinations, ABA concentrations increased progressively from stage 1 to stage 3, with a markedly stronger rise observed in Shurab3–Shahrud12 than in GN 15–Mamaee. From stage 3 to stage 5, ABA levels declined in both combinations, with a sharper reduction in Shurab3–Shahrud12 (Fig. 6A). The IAA concentration showed a continuous upward trend throughout all stages, with Shurab3–Shahrud12 consistently exhibiting higher IAA levels than GN15–Mamaee (Fig. 6B). Similarly, $GA_3$ content

**Table 6. Correlation matrix of morphological traits and photosynthetic pigments in studied rootstock-scion combinations.**

| | TH | NF | BD | NN | NI | IL | SL | NLB | SCSA | TC | TCSA | Chla | Chl b | TChl | Carotenoids |
|---|---|---|---|---|---|---|---|---|---|---|---|---|---|---|---|
| TH | 1 | | | | | | | | | | | | | | |
| NF | 0.5** | 1 | | | | | | | | | | | | | |
| BD | 0.3* | 0.93** | 1 | | | | | | | | | | | | |
| NN | 0.71** | 0.54** | 0.33** | 1 | | | | | | | | | | | |
| NI | 0.71** | 0.54** | 0.33** | 1** | 1 | | | | | | | | | | |
| IL | −0.01 ns | 0.26* | 0.21ns | −0.15 ns | −0.15 ns | 1 | | | | | | | | | |
| SL | 0.76** | 0.6** | 0.33** | 0.81** | 0.81** | 0.3* | 1 | | | | | | | | |
| NLB | −0.16 ns | 0.25* | 0.24ns | 0.06ns | 0.06ns | 0.48** | 0.26* | 1 | | | | | | | |
| SCSA | 0.36** | 0.29* | 0.18ns | 0.25* | 0.25 | 0.29* | 0.42** | 0.23ns | 1 | | | | | | |
| TC | 0.36** | 0.21ns | 0.06ns | 0.4** | 0.4** | 0.06ns | 0.43** | 0.04ns | 0.66** | 1 | | | | | |
| TCSA | 0.36** | 0.21ns | 0.06ns | 0.41** | 0.41** | 0.06ns | 0.43** | 0.04ns | 0.66** | 0.99** | 1 | | | | |
| Chl a | 0.28* | 0.3* | 0.32* | 0.39** | 0.39** | −0.26* | 0.16ns | −0.2 ns | 0.05ns | 0.13ns | 0.13ns | 1 | | | |
| Chl b | −0.52** | −0.43** | −0.26* | −0.39** | −0.39** | −0.2 ns | −0.59** | 0.04ns | −0.08 ns | −0.07 ns | −0.07 ns | 0.03ns | 1 | | |
| TChl | 0.54** | 0.41** | −0.24 ns | −0.35** | −0.35** | −0.2 ns | −0.55** | 0.06ns | −0.06 ns | −0.05 ns | −0.05 ns | 0.09ns | 0.99** | 1 | |
| Carotenoids | 0.18ns | 0.23ns | 0.32* | 0.16ns | 0.16ns | −0.22 ns | −0.03 ns | −0.42** | −0.2 ns | −0.1 ns | −0.1 ns | 0.56** | −0.01 ns | 0.03ns | 1 |

**, * and ns; significant at 0.01, 0.05 probability levels and not significant, respectively.

LF: Late Flowering, TH: Tree Height, NF: Number of Flowers per scaffold, BD: Blooming Density, SL: Scaffold length, NN: Number of Node per scaffold, NI: Number of internode, IL: Internode Length, NLB: Number of lateral Branches, TCSA: Trunk cross-sectional area, TC: Trunk Circumference, SCSA: scaffold cross-sectional area, Chl a: chlorophyll a, Chl b: chlorophyll b, TChl: Total chlorophyll.

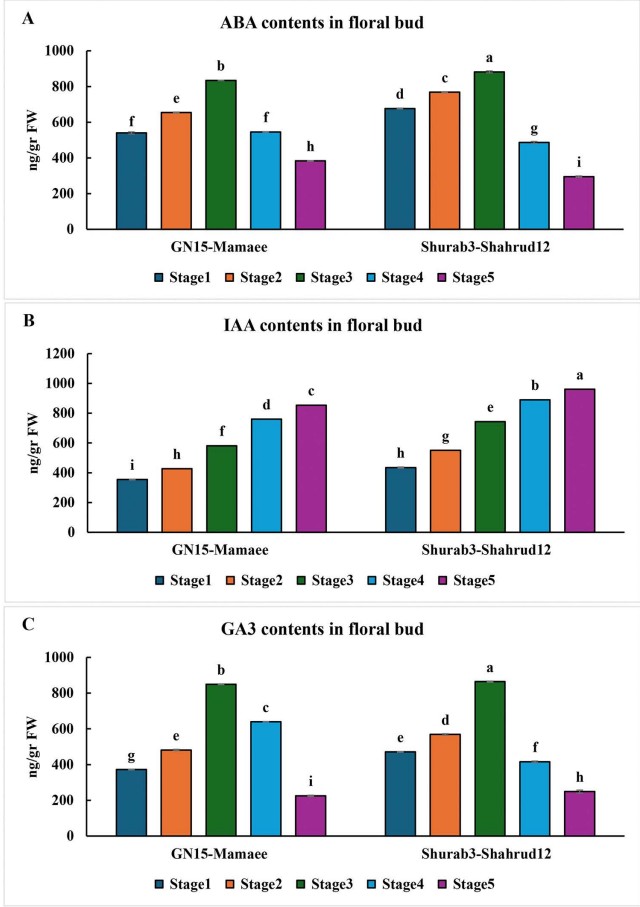

**Fig 6. Changes in ABA, IAA, and GA3 levels during floral bud differentiation in response to rootstock-scion combinations at five stages of development.** (A) ABA content in floral buds, (B) IAA content in floral buds and (C) GA3 content in floral buds. Columns with the same superscript letter are not significantly different (P < 0.05, LSD test).

increased from stage 1 to stage 3 in both combinations, followed by a decline toward stage 5. The increase and subsequent decrease were more pronounced in Shurab3–Shahrud12, suggesting a dynamic hormonal balance associated with enhanced floral induction (Fig. 6C).

## Expression patterns of flowering genes

The relative expression levels of flowering induction–related genes were examined in the rootstock–scion combinations Shurab3–Shahrud12 and Shurab3–Mamaee (with *Shurab3* as the control), as well as GN15–Shahrud12 and GN15–Mamaee (with GN15 as the control), using both leaf and floral bud samples (Supporting information). In the Shurab3–Shahrud12 combination, *AP1* expression was significantly higher than that of the control (Shurab3) at all developmental stages, except for bud stages 2 and 4. The Shurab3–Mamaee combination exhibited even greater *AP1* expression levels compared with Shurab3–Shahrud12 (Fig 7A). For the *CO* gene, expression in Shurab3–Shahrud12 remained consistently and significantly higher than in all other combinations across both leaf and floral bud stages (Fig 7B). Furthermore, *CO* transcript levels were generally higher in leaf samples than in floral bud samples, consistent with its known role in photoperiod-dependent flowering regulation.

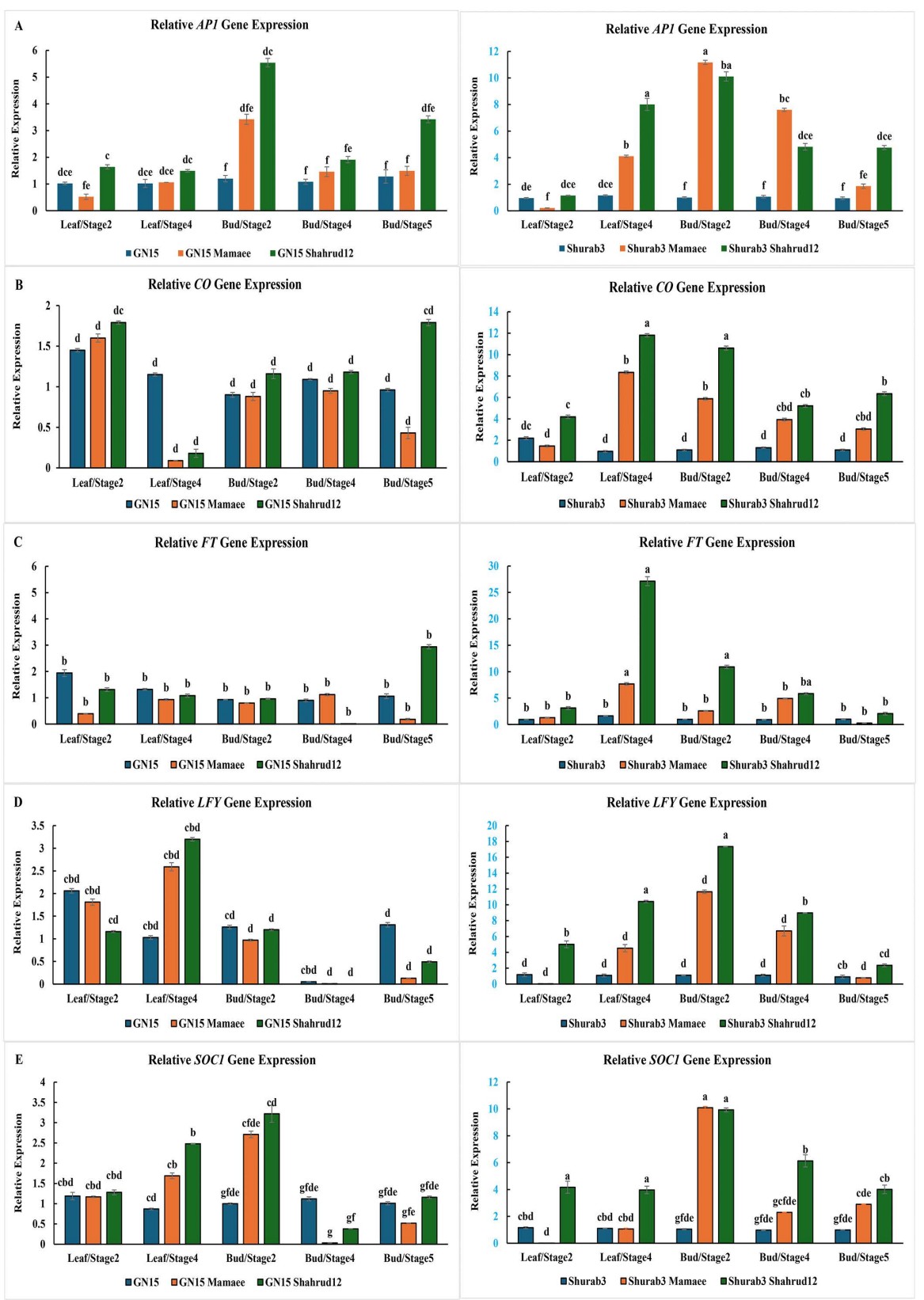

**Fig 7. Relative expression levels of genes related to flowering induction in leaf (Leaf/Stages 2,4) and floral bud (Bud/Stages 2,4,5) by RT-qPCR.** A: *AP1* expression, B: *CO* expression, C: *FT* expression, D: *LFY* expression and E: *SOC1* expression. Columns with the same superscript letter are not significantly different (P<0.05, LSD test).

The *FT* gene, a key regulator of floral induction predominantly expressed in leaves, displayed markedly elevated expression in the Shurab3–Shahrud12 combination compared with both its control (Shurab3) and the other graft combinations, reaching a peak at leaf stage 4. In contrast, the GN15–Mamaee combination showed reduced *FT* expression relative to its control (GN15) (Fig 7C). The *LFY* gene exhibited a similar expression pattern, with Shurab3–Shahrud12 showing higher transcript levels than both the control and other combinations. However, *LFY* expression declined sharply from bud stage 4 to bud stage 5 (Fig 7D). Expression of *SOC1* peaked at bud stage 2 in all combinations relative to their respective controls. At this stage, Shurab3–Mamaee exhibited the highest *SOC1* expression, slightly surpassing Shurab3–Shahrud12 and significantly exceeding the levels in other graft combinations (Fig 7E).

Overall, the gene expression profiles demonstrated that the enhanced flowering observed in Shurab3–Shahrud12 and, to a lesser extent, Shurab3–Mamaee was associated with upregulation of key flowering-related genes (*FT*, *CO*, *SOC1*, *LFY*, and *AP1*) in both leaves and floral buds. The coordinated elevation of *FT* and *CO* expression in leaves, together with the strong activation of *SOC1*, *LFY*, and *AP1* in floral buds, suggests that Shurab3-based grafts possess a more efficient signaling cascade linking photoperiod perception and floral meristem identity. These transcriptional trends were consistent with the hormonal profiles, where higher IAA and GA$_3$ levels and reduced ABA accumulation corresponded to stronger floral induction. Collectively, these results indicate that Shurab3 rootstocks promote flowering through the integrated regulation of hormonal balance and the activation of flowering pathway genes, ultimately leading to superior floral performance compared with GN15-based combinations.

## Discussion

Flowering induction represents a critical developmental transition in perennial fruit trees and is regulated by a complex interplay between environmental cues and endogenous molecular and hormonal networks [17,29]. Numerous studies have sought to elucidate the mechanisms underlying this process across different species [24]. Endogenous factors, including abscisic acid (ABA) [47–49], fluctuations in carbohydrate and nutrient metabolism, elevated gibberellin (GA) levels [48,50–52], and fruit load [41,53], are known to influence both the timing and intensity of flowering. In commercial almond cultivars, the initiation and differentiation of floral buds constitute key phases of the reproductive cycle, directly impacting productivity and overall orchard performance.

Rootstocks, as integral components of grafted systems, can markedly influence the physiological, morphological, and reproductive traits of the scion [2,4,12,22,54–57]. Our results demonstrate that rootstock–scion interactions significantly modulate almond flowering by coordinating physiological, hormonal, and molecular processes. In particular, the Shurab3–Shahrud12 combination exhibited substantially enhanced flowering rates and flower bud formation compared with the other combinations. This superior performance likely reflects the ability of the Shurab3 rootstock to optimize nutrient translocation and maintain hormonal balance in apical buds, thereby promoting floral meristem initiation. In contrast, the GN15–Mamaee graft displayed reduced floral abundance, which may result from intrinsic limitations in assimilate distribution and hormone regulation.

Recent advances have further emphasized that rootstock–scion interactions influence flowering not only through nutrient and hormone translocation but also via small signaling molecules such as microRNAs and peptides. Hayat et al. (2023) demonstrated that root-derived hormonal ratios, particularly ABA/IAA/GA balance, play a crucial role in transmitting developmental cues to the scion, thereby modulating the transition from vegetative to reproductive phases in fruit trees [58]. Similarly, Tedesco et al. (2023) highlighted that compatible rootstock–scion combinations enhance the bidirectional

exchange of metabolites, RNAs, and phytohormones, ultimately improving floral induction and fruiting performance [59]. These findings are consistent with our results, suggesting that the superior flowering observed in the Shurab3–Shahrud12 combination may result from more efficient long-distance communication and hormonal homeostasis between root and shoot tissues. Consequently, optimizing rootstock selection can serve as a promising physiological and molecular strategy for enhancing flowering efficiency and yield potential in commercial almond production systems.

These findings are consistent with previous studies highlighting the critical role of rootstocks in shaping both reproductive and agronomic performance. For example, Cong et al. (2023) reported that grafting the pear cultivar Abbé Fetel onto the 'Yunnan' quince rootstock significantly enhanced flowering rates, demonstrating the rootstock's capacity to promote floral induction [60]. Similarly, Pica et al. (2021) showed that different clonal rootstocks (GF677, Rootpac® 20, Rootpac® R) differentially affected flowering time and vegetative vigor in the Spanish almond cultivar Guara [7]. Lordan et al. (2019) further confirmed that a diverse range of rootstocks, including Cadaman®, Garnem®, INRA GF-677, IRTA-1, IRTA-2, Ishtara®, Adesoto, Rootpac® 20, Rootpac® 40, and Rootpac® R,significantly influenced the horticultural performance of the Marinada and Vairo cultivars [10]. Collectively, these studies reinforce that rootstock selection not only regulates vegetative vigor but also modulates reproductive traits, including flowering dynamics and cultivar adaptability.

Principal component analysis (PCA) indicated that four components accounted for 89.64% of the total variance in morphological and photosynthetic traits among the tested rootstock–scion combinations. This finding aligns with Akin et al. (2022) [61], who demonstrated the utility of PCA in elucidating complex trait relationships in fruit crops. In the present study, PC1 was primarily associated with structural traits such as trunk cross-sectional area (TCSA) and tree height, whereas PC2 was linked to photosynthetic pigments, including chlorophyll $a$ and chlorophyll $b$. These associations highlight the distinct influence of rootstock–scion interactions on overall plant performance. The Shurab3–Shahrud12 combination exhibited high PC3 scores, reflecting superior floral induction and reproductive performance, while GN15–Mamaee displayed negative PC1 and PC2 scores, indicative of reduced vigor and pigment content. These clustering patterns underscore the potential of Shurab3 as a promising rootstock for enhancing yield-related traits in commercial almond orchards.

Plant hormones are central regulators of flowering, influencing not only floral initiation but also vegetative growth, apical dominance, fruit development, and responses to abiotic stress [13,16,62,63]. By coordinating these processes, hormones play a pivotal role in plant growth and development, with particular importance in floral induction [64,65]. Modifications in the hormonal composition of apical buds critically affect flower bud differentiation, and rootstocks have been shown to modulate hormone levels in the scion [4,54–56]. The present study, which quantified hormone concentrations in floral buds across five developmental stages, highlights the regulatory role of rootstocks in shaping hormonal dynamics during flowering. In both rootstock–scion combinations, ABA concentrations increased from stage 1 to stage 3 and declined thereafter toward stage 5, a trend consistent with previous reports in sweet cherry, pear, and persimmon [60,62,66]. IAA levels rose steadily from stage 1 to stage 5, corroborating findings in sweet cherry where IAA promotes floral initiation [62,66–68]. Gibberellins, particularly $GA_3$, are well-established regulators of major flowering pathways [66,69,70]. In the current study, $GA_3$ content increased from stage 1 to stage 3, more markedly in Shurab3–Shahrud12, and subsequently declined, aligning with observations in previous studies [60,66,71]. Collectively, these results suggest that an optimal hormonal balance, characterized by moderate ABA accumulation, sustained IAA increase, and transient $GA_3$ elevation, is essential for efficient floral induction.

The observed differences in flowering among rootstock–scion combinations were accompanied by distinct physiological and biochemical profiles. Combinations exhibiting superior flowering, particularly Shurab3–Shahrud12, also displayed higher photosynthetic pigment concentrations, including chlorophyll a, chlorophyll b, total chlorophyll, and carotenoids. These elevated pigment levels likely enhance photosynthetic efficiency, providing greater assimilate availability to support floral induction and bud development.

Recent research further underscores the importance of carbohydrate partitioning and stress-signal regulation as critical components of rootstock-mediated flowering responses. For instance, in mango (Mangifera indica) scions grafted onto different rootstocks, Vittal et al. (2023) demonstrated that rootstocks altered carbohydrate metabolism and nutrient content, thereby influencing scion physiology and flowering traits [72]. Moreover, in rootstock–scion combinations of 'Cuiguan' pear, Liang et al. (2024) showed that winter-dormant carbohydrate reserves are strongly correlated with spring bud break and flowering intensity, indicating that assimilate availability preceding floral induction is a key variable [73]. In addition, a detailed study in apple (Malus × domestica) rootstock systems found that rootstocks influenced leaf isotope composition and carbon assimilation/partitioning pathways under different water and nutrient regimes [74]. Together, these findings support the idea that rootstocks do not merely provide passive support but actively regulate source-sink dynamics and stress resilience of the scion, factors which can feed into floral-meristem initiation and bud differentiation. In light of our findings, where superior flowering in the Shurab3–Shahrud12 combination was accompanied by higher pigment levels, better photosynthetic indicators, and improved hormonal profiles, it is plausible that the rootstock also enhanced carbohydrate allocation and stress-tolerance capacity in the scion, thereby creating a more favourable internal environment for floral induction.

Hormonal analyses further revealed that higher levels of IAA and $GA_3$, coupled with reduced ABA accumulation, were associated with increased flower bud formation in the high-performing grafts. These hormonal profiles suggest that rootstock-mediated modulation of the auxin–gibberellin–ABA balance plays a pivotal role in regulating floral meristem initiation and subsequent flower development.

At the transcriptomic level, recent studies have shown that rootstocks exert wide-ranging effects on gene expression in the scion. He et al. (2022) reported that grafting significantly alters the transcriptional regulation of genes associated with carbohydrate transport, hormone signaling, and floral meristem identity in several fruit tree species [75]. In almonds, Montesinos et al. (2023) observed differential expression of auxin-, gibberellin-, and abscisic acid-related genes between contrasting rootstock–scion combinations, reinforcing the concept that rootstocks modulate key molecular networks governing reproductive development [3]. In line with these reports, the enhanced expression of *FT*, *SOC1*, *LFY*, and *AP1* observed in the Shurab3–Shahrud12 graft suggests that this rootstock may promote floral induction by coordinating hormonal signaling with the activation of major flowering pathways.

At the molecular level, expression patterns of key flowering-related genes corroborated these physiological and hormonal observations. Rootstocks can modulate flowering by influencing the transcriptional regulation of key genes in the scion [5,12,17,60,76,77]. Among the central regulators, *FT*, *CO*, *SOC1*, *LFY*, and *AP1* constitute the core flowering network [17,46,78–81]. *FT*, activated by the transcription factor *CO*, produces a florigen signal that is transported to the shoot apical meristem, where it induces *AP1* and *SOC1*. These genes, in turn, promote *LFY* expression, collectively driving the vegetative-to-reproductive transition [29]. In the present study, *AP1* expression was significantly higher in rootstock–scion combinations than in controls across all developmental stages, becoming detectable following floral meristem initiation, consistent with observations by William et al. (2004) and Cong et al. (2023) [60,82]. *CO* expression increased during leaf development but declined in floral buds, reflecting its established role as a photoperiodic regulator [29,83]. *FT* expression was notably elevated in leaves, particularly at the fourth leaf stage, confirming its function as a systemic floral inducer [29,84–87]. *LFY* expression peaked during early floral bud formation and decreased subsequently, in line with its role as a master regulator of meristem identity [29,88–90]. Similarly, *SOC1*, which regulates *LFY*, exhibited a comparable pattern and was detected in multiple tissues, including roots, shoots, and inflorescences [90,91]. Together, *SOC1* and *FT* activate *LFY* and *AP1*, orchestrating floral meristem initiation at the shoot apex [28,81,92,93].

Collectively, the hormonal and molecular data demonstrate that the Shurab3 rootstock, when grafted with either Mamaee or Shahrud12, enhances both the expression of key flowering genes and the production of growth-promoting hormones during floral induction. This coordinated regulation results in increased flower induction and a higher number of floral buds. The novelty of the present study lies in elucidating, for the first time, the combined effects of rootstock-mediated hormonal modulation and gene expression dynamics on floral induction in almond. These findings

offer valuable insights for optimizing rootstock selection and orchard management strategies to improve yield stability and reproductive performance under varying environmental conditions.

## Conclusion

Rootstock selection strongly influences floral induction and flower bud formation in commercial almond cultivars. Among the rootstocks tested, Shurab3 consistently enhanced flowering, particularly with Shahrud12, while GN15 showed the weakest performance. The superior effect of Shurab3 across both cultivars highlights the critical role of rootstock–scion interactions in regulating reproductive success. These results support the use of Shurab3 as a flower-inducing rootstock in environments similar to the study site and provide a foundation for further research into the molecular and physiological mechanisms underlying rootstock-mediated flowering regulation.

## Supporting information

**S1 Table. Component weights of morphological and physiological traits for the first four principal components (PC1–PC4) derived from principal component analysis.**
(DOCX)

**S1 File. Morphological and photosynthetic pigment data, along with their statistical analyses.**
(XLSX)

**S2 File. Plant hormone data and the results of their statistical analyses.**
(XLSX)

**S3 File. The relative expression levels of flowering-related genes together with their corresponding analyses performed in this study.**
(XLSX)

## Acknowledgment

This research was partially supported by Shahrekord University and Iran National Science Foundation (INSF). We would like to acknowledge the Biotechnology Research Institute of Shahrekord University that support our research on laboratory experiments. We are grateful to Dr. S. H. Nourbakhsh who provided valuable almond samples.

## Author contributions

**Conceptualization:** Behrouz Shiran, Abdolrahman Mohamadkhani, Habibollah Nourbakhsh.

**Investigation:** Masoud Abedian-Chermahini.

**Methodology:** Masoud Abedian-Chermahini.

**Resources:** Habibollah Nourbakhsh.

**Supervision:** Behrouz Shiran, Abdolrahman Mohamadkhani, Habibollah Nourbakhsh.

**Validation:** Masoud Abedian-Chermahini.

**Writing – original draft:** Masoud Abedian-Chermahini.

**Writing – review & editing:** Behrouz Shiran, Abdolrahman Mohamadkhani, Habibollah Nourbakhsh.

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
