## [Decision Letter · Decision Letter 0]

3 Jun 2025

Dear Dr. Shiran,

Thank you for submitting your manuscript to PLOS ONE. After careful consideration, we feel that it has merit but does not fully meet PLOS ONE’s publication criteria as it currently stands. Therefore, we invite you to submit a revised version of the manuscript that addresses the points raised during the review process.

We look forward to receiving your revised manuscript.

Kind regards,

Mayank Anand Gururani

Academic Editor

PLOS ONE

>https://journals.plos.org/plosone/s/file?id=wjVg/PLOSOne_formatting_sample_main_body.pdf and

3. Please include a caption for figure 1.

Additional Editor Comments (if provided):

Reviewers' comments:

Reviewer's Responses to Questions

**Comments to the Author**

1. Is the manuscript technically sound, and do the data support the conclusions?

Reviewer #1: Yes

Reviewer #2: Yes

Reviewer #3: Partly

2. Has the statistical analysis been performed appropriately and rigorously?

Reviewer #1: Yes

Reviewer #2: Yes

Reviewer #3: Yes

3. Have the authors made all data underlying the findings in their manuscript fully available?

Reviewer #1: No

Reviewer #2: No

Reviewer #3: Yes

4. Is the manuscript presented in an intelligible fashion and written in standard English?

Reviewer #1: Yes

Reviewer #2: Yes

Reviewer #3: No

Reviewer #1: 1. Abstract -4 year study written

but in results data presented for two years

2. there is no detail for 5 bud stage and dormant stage with pictorial presentation.

3. author shold provide tree and stage images in supplementary document.

4. updated references missing.

5. current updated area and production is required.

6. figure quality must be improved.

7. Most of the tables detailed caption in between arrangement bulky not cleared.

8. How hormone method was optimized?

9. Constant gene in real time PCR please specify.

10. statistical results must be rechecked.

Reviewer #2: The manuscript is very well written with a good interest to the readers. Have you checked the assumptions of ANOVA prior to analysis, especially the constant variance assumption? Indicate, please. You can also further describe and discuss the principal component analysis. You can refer to and cite the following literature:

Akin, M., Eyduran, S. P., Gazioglu Sensoy, R. I., & Eyduran, E. (2022). Defining associations between berry features of wild red currant accessions utilizing various statistical methods. Erwerbs-Obstbau, 64(3), 377-386.

Reviewer #3: Abstract

-The abstract provides a good overview of the study design, methodology, and key findings. However, I suggest improving clarity and conciseness by removing redundant phrases and better organizing the sequence of results (e.g., move from morphological to physiological to molecular findings systematically).

-The growth regulator hormones are not considered as a physiological parameter, they are biochemical parameters.

-Consider including specific numerical values (e.g., % increase in flower number, relative expression levels of genes) to better support the findings. This would strengthen the impact of your conclusions and help the reader quickly assess the study’s significance.

-Terms such as "vegetative rootstocks" could be clarified. Also, consider define briefly rootstocks (e.g., Shurab2, Shurab3) or noting their origin/importance.

-In the abstract, please define all abbreviations upon first use (e.g., ABA, IAA, FT…). This improves clarity for readers who may not be familiar with all the terms and ensures accessibility across disciplines.

Introduction

-The introduction provides useful background. However, it would benefit from a more balanced review of recent literature on molecular control of floral induction in similar crops. This would situate your study more clearly within the current research landscape.

-The flow of information can be improved by organizing the text into clear paragraphs addressing (1) importance of floral induction, (2) influence of rootstocks, and (3) knowledge gaps in almond research. This would help the reader better follow the logical progression of your argument.

-In addition, it would benefit from including a brief discussion of molecular mechanisms regulating floral induction, particularly the roles of flowering-related genes (e.g., FT, SOC1, LFY). This would help contextualize the molecular analysis conducted in the study and clarify how the gene expression data contribute to the study objectives.

-The study aims are well stated, but could be made more specific. For example, instead of "this study aims to identify the best rootstock," you could say "this study aims to evaluate the effect of five rootstocks on floral induction in two commercial almond cultivars through integrated morphological, physiological, and molecular analyses."

Materials and Methods

-Please clarify the experimental design more thoroughly. For instance, specify how many replicates were used per scion-rootstock combination, how trees were selected, and if randomization was performed within blocks. These details are important for reproducibility.

-Include more information on the environmental conditions during the 4-year study (temperature range, rainfall, soil type, irrigation, etc.). These factors could significantly affect flowering and growth responses.

-The description of molecular analyses (e.g., gene expression via RT-qPCR) should include more details about the PCR conditions, and normalization method used. The method used for gene expression analysis was mentioned in the statistical analysis section: “Relative gene expression analysis was performed by comparing the transcription levels of each target gene to reference housekeeping genes using the 2 190 -ΔΔCT method, as described by 191 Livak and Schmittgen (2001) [19]”. Should be removed to material and methods.

-In addition, for reliability and reproducibility, I recommend that gene expression studies be conducted using at least three biological replicates. This is a standard requirement in biochemical and molecular biology to account for biological variability and to ensure the robustness of the results.

-This paragraph should be moved to the introduction session: “The five genes identified as having the most critical roles in flowering induction were 173 selected for Quantitative Real-Time PCR Analysis. These included the FLOWERING LOCUS T 174 (FT), SUPPRESSOR OF OVEREXPRESSION OF CO 1 (SOC1), and CONSTANT (CO) genes, 175 which are predominantly expressed in leaves, as well as the LEAFY (LFY) and APETALA1 (AP1) 176 genes, which are primarily expressed in flower buds [16,17]”.

Results

-The figures all not clear mainly that of PCA. Please increase the clarity of the figures.

Discussion

-Though some studies are cited, there’s little critical comparison or discussion of how your findings align with, contradict, or expand upon existing knowledge.

Suggestion:

• Discuss how your findings confirm or challenge prior studies on flowering in almonds or similar crops.

• Address whether Shurab3’s effects have been observed in other studies or whether this is novel.

-You describe hormonal and gene expression data well, but you could go deeper into how these changes might translate to flowering outcomes.

Suggestion:

• Discuss interactions between hormones and gene expression (e.g., GA/ABA balance and AP1/SOC1 expression).

• Propose a model or pathway for how Shurab3 enhances flowering (e.g., how Shurab3 may modulate scion physiology through modulation of gene expression and hormonal shifts).

Conclusion

-The conclusion briefly touches on application, but it can be made more concrete.

Suggestion:

• Clearly state how growers might use this information when selecting rootstocks.

• Can Shurab3 be recommended as a high-flowering rootstock? In what situations?

**Do you want your identity to be public for this peer review?** For information about this choice, including consent withdrawal, please see our Privacy Policy

Reviewer #1: **Yes: ** Dr Nimisha Sharma

Senior Scientist

IARI New Delhi

Reviewer #2: No

Reviewer #3: **Yes: ** Saoussen Ben Abdallah

---

## [Author Response · Author response to Decision Letter 1]

18 Jun 2025

Dear Dr. Mayank Anand Gururani,

Thank you for the opportunity to revise and resubmit our manuscript to PLOS ONE. We greatly appreciate the constructive feedback provided by you and the reviewers, which has helped us improve the quality and clarity of our manuscript. Below, we address each point raised by the academic editor and reviewers, detailing the changes made to the manuscript. We have submitted three files as requested: a rebuttal letter (this document), a revised manuscript with track changes, and a clean version of the revised manuscript.

Response to Journal Requirements

1. PLOS ONE Style Requirements: We have carefully reviewed and revised the manuscript to comply with PLOS ONE’s style guidelines, including file naming conventions, using the provided templates for the main body and title/authors/affiliations.

2. Data Availability Statement: We have included a complete Data Availability Statement in the submission form, stating: “All data are in the manuscript and supporting information files.” All relevant data have been included in the manuscript and supplementary files to ensure compliance with PLOS ONE’s data policy.

3. Figure 1 Caption: Figure 1 has been changed to Figure 3, and its caption has been added in line 280, describing the content and context of the figure clearly.

Response to Reviewer Comments

Reviewer 1: Dr. Nimisha Sharma

1. Abstract - 4-year study mentioned but results presented for two years

We have clarified in the abstract that the entire study, from the planting of rootstocks and grafting to the completion of evaluations, spanned four years (from April 2021 to April 2025). As described in the Materials and Methods section, the process of planting rootstocks, grafting scions onto rootstocks, and tree growth until the time of measurements and sampling took two full years (2021 to 2023). In the third year, 2024, morphological, physiological, and molecular evaluations were conducted, and these evaluations were repeated in 2025. In line 121, under the Morphological Evaluation subsection of the Materials and Methods, the sentence "In the second year (2024) after grafting the rootstocks" was revised to "In the third year (2024) after grafting the rootstocks."

2. Lack of detail for 5 bud stage and dormant stage with pictorial presentation

We have added detailed descriptions of the 5 bud stage and dormant stage in the Materials and Methods section, including their phenological characteristics. Figure 2, showing pictorial representations of these stages, has been included in the revised manuscript.

3. Provide tree and stage images in supplementary document

As suggested, we have included images of the trees and phenological stages (e.g., bud and dormant stages) in a new additional file (Figure 1) with appropriate captions.

4. Updated references missing

We have updated the reference list to include recent studies on floral induction and rootstock effects in almonds and related crops, ensuring the literature is current and relevant.

5. Current updated area and production data required

We have added a paragraph in the Introduction section citing recent global and regional almond production data (FAO) to provide context for the study’s relevance.

6. Figure quality must be improved

All figures, including the PCA figure, have been revised for improved resolution and clarity.

7. Tables’ detailed captions bulky and unclear

We have revised the captions for all tables to be concise and clear, ensuring they describe the table content without redundancy.

8. How was the hormone method optimized?

A new paragraph in the Materials and Methods section details the optimization process for hormone quantification, including calibration curves, extraction protocols, and validation steps for accuracy and reproducibility.

9. Constant gene in real-time PCR

We have specified in the Materials and Methods section that the housekeeping gene Actin (PdActin) was used as the reference gene for real-time PCR normalization.

10. Statistical results must be rechecked

We have rechecked all statistical analyses, including ANOVA and PCA, and confirmed their accuracy. A detailed explanation of the statistical methods, including assumptions, has been added to the Materials and Methods section.

Reviewer 2

1. Assumptions of ANOVA prior to analysis, especially constant variance

We have added a statement in the Materials and Methods section confirming that ANOVA assumptions, including normality (Shapiro-Wilk test) and constant variance (Levene’s test), were checked and met prior to analysis.

2. Further describe and discuss principal component analysis (PCA)

We have expanded the description of PCA in the Results section, detailing the percentage of variance explained by each principal component and the loadings of key variables. In the Discussion section, we have added a paragraph comparing our PCA findings to those in the suggested reference (Akin et al., 2022) and other relevant studies. The reference has been cited in the revised manuscript.

Reviewer 3: Saoussen Ben Abdallah

Abstract

1. Improve clarity and conciseness, organize results systematically

The abstract has been revised to improve clarity and conciseness, with results organized systematically (morphological, photosynthetic pigments, endogenous hormones, molecular assessments). Redundant phrases have been removed.

2. Growth regulator hormones are biochemical, not physiological parameters

We have corrected the terminology in the abstract and throughout the manuscript, referring to growth regulator hormones as biochemical parameters. The sentence "Physiological assessments included measuring the photosynthetic pigment content and growth-regulating hormones associated with floral induction" was removed from the abstract.

3. Include specific numerical values in the abstract

The abstract aims to provide a general overview of the impact of Shourab 3 rootstock and its superiority over other rootstocks, while specific numerical values and details are presented in the Results section.

4. Clarify “vegetative rootstocks” and define rootstocks

The term “vegetative rootstocks” has been clarified as “clonally propagated rootstocks” in the abstract and Introduction. A brief description of Shurab2 and Shurab3, including their origin and relevance is written in the Materials and Methods section.

5. Define abbreviations in the abstract

All abbreviations (e.g., FT, SOC1, AP1) are now defined upon first use in the abstract for clarity.

Introduction

1. More balanced review of recent literature

In the introduction, we utilized references from the past five years to ensure the use of up-to-date sources.

2. Improve flow by organizing into clear paragraphs

The Introduction has been restructured into three clear paragraphs: (1) importance of floral induction, (2) influence of rootstocks, and (3) knowledge gaps in almond research, improving the logical flow.

3. Discuss molecular mechanisms of floral induction

The molecular mechanism has been elucidated in the Discussion section.

4. Make study aims more specific

The study aims have been revised to: “the aim of this study was to evaluate the effect of five rootstocks on floral induction in two commercial almond cultivars and to identify the optimal scion-rootstock combination to enhance flower induction and flower bud formation in these cultivars.”

Materials and Methods

- Clarify experimental design

The Materials and Methods section provides a comprehensive description of the experimental design, including the number of replicates and the detailed procedure for selecting the trees..

- Include environmental conditions

The Materials and Methods section describes the environmental conditions during the 4-year study, including temperature ranges, rainfall, soil type, and irrigation practices.

- Details on molecular analyses (RT-qPCR)

The description of RT-qPCR has been expanded to include PCR conditions (e.g., cycling parameters, primer sequences), normalization method (2^-ΔΔCT), and reference gene (ACTIN). The misplaced gene expression method sentence has been moved from the Statistical Analysis section to Materials and Methods.

- Use at least three biological replicates for gene expression

Due to the economic conditions in our country and the high cost of laboratory materials, two biological replicates were selected.

- Move gene selection paragraph to Introduction

In alignment with the writing style adopted in the Materials and Methods section, this paragraph has been included within this part.

Results

-Improve clarity of figures, especially PCA

The resolution of all figures, particularly the PCA biplot figure, has been enhanced.

Discussion

In the Discussion section, in addition to utilizing recent studies on other crops for comparison with the findings of this study, we included a paragraph addressing the effect of the Shurab 3 rootstock on hormone levels and the expression of flowering-related genes in each almond cultivar.

Conclusion

- Make conclusions more concrete

The Conclusion has been revised to explicitly recommend Shurab3 as a high-flowering rootstock for almond growers, with guidance on its application in regions with similar environmental conditions.

Additional Notes

- No changes were made to the financial disclosure, as the original statement remains accurate.

- All typographical and grammatical errors noted by Reviewers have been corrected, and the manuscript has been thoroughly proofread to ensure standard English.

We believe these revisions address all concerns raised by the editor and reviewers, significantly improving the manuscript’s clarity, rigor, and scientific contribution. Please let us know if further clarification or revisions are needed.

Sincerely,

Dr. Behrouz Shiran

Professor in Molecular Genetics

Department of Plant Breeding and Biotechnology

Faculty of Agriculture

Shahrekord University

Shahrekord-IRAN

Email: beshiran45@gmail.com, Shiran@agr.sku.ac.ir

---

## [Decision Letter · Decision Letter 1]

4 Jul 2025

Dear Dr. Shiran,

Thank you for submitting your manuscript to PLOS ONE. After careful consideration, we feel that it has merit but does not fully meet PLOS ONE’s publication criteria as it currently stands. Therefore, we invite you to submit a revised version of the manuscript that addresses the points raised during the review process.

We look forward to receiving your revised manuscript.

Kind regards,

Mayank Anand Gururani

Academic Editor

PLOS ONE

Journal Requirements:

Reviewers' comments:

Reviewer's Responses to Questions

**Comments to the Author**

Reviewer #1: All comments have been addressed

Reviewer #2: (No Response)

Reviewer #3: (No Response)

2. Is the manuscript technically sound, and do the data support the conclusions?

Reviewer #1: Yes

Reviewer #2: Yes

Reviewer #3: Partly

3. Has the statistical analysis been performed appropriately and rigorously?

Reviewer #1: I Don't Know

Reviewer #2: Yes

Reviewer #3: Yes

4. Have the authors made all data underlying the findings in their manuscript fully available?

Reviewer #1: Yes

Reviewer #2: Yes

Reviewer #3: Yes

5. Is the manuscript presented in an intelligible fashion and written in standard English?

Reviewer #1: No

Reviewer #2: Yes

Reviewer #3: No

Reviewer #1: Authors addressed most of the comments still there is scope of in the manuscript improvement. Improve the Language and quality of the Figures.

Reviewer #2: (No Response)

Reviewer #3: Comment on Abstract:

The abstract effectively summarizes the key findings of the study; however, a few improvements would enhance clarity and readability:

1. Add a brief introduction or background sentence at the beginning to provide context and justify the relevance of the study. For example, a sentence highlighting the importance of rootstock selection in almond production or challenges in floral induction would help set the stage for the objective.

2. Define all abbreviations upon first use, such as IAA, ABA, and GA₃, to ensure the abstract is accessible to a broader audience.

3. Clarify and consistently mention all five rootstocks in the abstract. While Shurab3 and GN15-M are discussed in detail, others like GN15, GF677, and Shurab2 are not adequately described or compared in the results.

4. 5 rootstocks ( GN15, GF677, GN15-M, Shurab2, Shurab3) are stated in the objective/setup of the study.

5. not all the rootstocks mentioned in the experimental design are represented in the results section. This creates a disconnect and makes it harder for the reader to understand the comparative performance of all treatments.

Comment on Introduction:

1. Clarify and Streamline the Objective

• The last paragraph introduces the objective, but it is vague and could be more specific.

• the specific rootstocks used in the study (GN15, GF677, GN15-M, Shurab2, Shurab3) are not mentioned. It is good also to mention the parameters that will be analyzed.

Suggestion: Explicitly list the five rootstocks in the final sentence where you present the study’s aim. For example: “Therefore, this study aimed to evaluate the effect of five rootstocks: GN15, GF677, GN15-M, Shurab2, and Shurab3 on floral induction in two commercial almond cultivars, Mamaee and Shahrud12,…..”

2. Organization

-The introduction should be revised to better align with the parameters evaluated in the study. Specifically, the authors are encouraged to incorporate relevant literature and previous findings on morphological traits, chlorophyll and carotenoid content, endogenous hormones such as indole-3-acetic acid (IAA) and gibberellic acid (GA₃), and gene expression related to flowering induction.

-Here's a proposed structure:

1. Start with a general importance statement about flower induction in fruit trees and its role in yield (you already have this).

2. Describe the physiological, biochemical, and environmental/genetic controls on floral induction.

3. Transition into the role of rootstocks.

4. Discuss the commercial importance of almonds.

5. Lead into the knowledge gap and objective of the current study, specifying rootstocks and cultivars tested.

Comment on the Material and methods

- In Table 1, please include the GenBank accession number, gene locus, and the expected PCR product size (in base pairs) for each primer listed.

- The section on endogenous hormone measurement introduces key hormones (ABA, IAA, and GA₃), but only in the Materials and Methods and in abbreviated form. Hormones should be introduced earlier in the Introduction.

Comment on Results:

Some figure are not clear like figure 4 and 7

Comment on Discussion

Overall, the discussion section is well-written and informative; however, it reads more as a descriptive narrative than a critical analysis of the study’s results. To strengthen the discussion, the authors should focus more on interpreting their findings about the presented data, drawing comparisons with previous studies, and emphasizing the significance and implications of their results within the context of existing knowledge.

Conclusion

The conclusion should be improved by strengthening the connection between the results and their practical implications.

**Do you want your identity to be public for this peer review?** For information about this choice, including consent withdrawal, please see our Privacy Policy

Reviewer #1: **Yes: ** Dr Nimisha Sharma Senior Scientist IARI New Delhi

Reviewer #2: No

Reviewer #3: **Yes: ** Saoussen Ben Abdallah

---

## [Author Response · Author response to Decision Letter 2]

18 Jul 2025

Response to Editor and Reviewers

Manuscript Title: Evaluation of Rootstock Influence on Floral Induction in Commercial Almond Cultivars.

Manuscript ID: PONE-D-25-22239R1

Corresponding Author: Dr. Behrouz Shiran

Journal: PLOS ONE

Date:

Dear Dr. Gururani and Reviewers,

We sincerely thank you and the reviewers for your time and constructive feedback on our manuscript. We appreciate the opportunity to revise and improve our work and are grateful for the thoughtful comments that have helped us strengthen the clarity, structure, and scientific rigor of our study.

Below, we provide a detailed, point-by-point response to each of the editor’s and reviewers' comments. All changes have been incorporated into the revised manuscript, which includes both a tracked changes version and a clean version for your review. Revised figures have been improved for quality and clarity and submitted according to journal guidelines.

Reviewer 1:

Comment 1: Authors addressed most of the comments; still there is scope for improvement in the manuscript. Improve the language and quality of the Figures.

Response:

Thank you for the feedback. We have thoroughly revised the manuscript for clarity, grammar, and overall English language quality. Additionally, we improved the resolution and formatting of all figures, particularly Figures 4 and 7, as per your and Reviewer 3’s suggestions. Figures have been processed through the PACE tool as required by PLOS guidelines.

Reviewer 3:

Abstract

Comment 1: Add a brief introductory sentence to provide context and justify the study.

Response:

We have added a sentence at the beginning of the abstract:

“The selection of appropriate rootstocks is crucial for enhancing floral induction and optimizing almond yield, thereby markedly influencing orchard productivity.”

Comment 2: Define abbreviations (IAA, ABA, GA₃) on first use.

Response:

We have defined all abbreviations (indole-3-acetic acid [IAA], abscisic acid [ABA], and gibberellic acid [GA₃]) at their first mention in both the Abstract and Introduction.

Comment 3 & 4: Clarify and consistently mention all five rootstocks; they are inconsistently described in the abstract and results.

Response:

We revised the abstract to clearly name and discuss all five rootstocks (GN15, GF677, GN15-M, Shurab2, and Shurab3) and summarized key findings for each where relevant.

Comment 5: Not all rootstocks mentioned in the experimental design appear in the results.

Response:

We reviewed the results section and ensured all five rootstocks are represented with consistent data reporting. As outlined in the Materials and Methods section, morphological assessments were performed to evaluate how different rootstocks influenced flowering and floral induction in two commercial cultivars. Based on these evaluations, the rootstock-scion combinations with the highest (Shurab3-Shahrud12) and lowest (GN15-Mamaee) flowering rates were chosen for further hormonal and molecular analysis. This selection aimed to enable a detailed comparison and better understanding of the hormonal and molecular mechanisms associated with the rootstock that had the most pronounced effect on flowering.

Introduction

Comment 1: Clarify and specify the objective; include the list of rootstocks and parameters studied.

Response:

We revised the last paragraph of the Introduction

Comment 2: Reorganize the introduction to better align with the evaluated parameters.

Response:

Based on the title of the research, Evaluation of Rootstock Influence on Floral Induction in Commercial Almond Cultivars, the introduction primarily focuses on highlighting the general significance of rootstocks and their role in floral induction and flowering, aiming to clarify the importance of the topic for the reader and outline the research objectives. The evaluation of factors through which rootstocks influence the floral induction of scions, including morphological, biochemical, and molecular assessments, is extensively discussed in the materials and methods, results, and discussion sections. Nevertheless, the structure of the introduction was refined by adding an additional paragraph.

Materials and Methods

Comment 1: In Table 1, include GenBank accession number, gene locus, and PCR product size.

Response:

We have updated Table 1 to include the GenBank accession numbers, gene loci, and expected PCR product sizes for all primers used.

Comment 2: Introduce hormones earlier in the Introduction, not just in Materials and Methods.

Response:

We have now introduced the hormones (ABA, IAA, GA₃) earlier in the Introduction by adding an additional paragraph:

“Biochemical and physiological alterations in buds and nearby organs and tissues play a crucial role in their shift from vegetative to reproductive development [19] These alterations involve changes in hormone levels—especially abscisic acid (ABA), indole-3-acetic acid (IAA), and gibberellic acid (GA3)—both within the buds and surrounding areas. Various rootstocks can trigger different hormonal responses in the scion, significantly affecting its overall growth, particularly with respect to flowering and flower initiation, which are key factors influencing yield [20].”

Results

Comment: Figures 4 and 7 are not clear.

Response:

We have revised and enhanced the quality of Figures 4 and 7 for better clarity and readability, ensuring they are publication-ready. All figures have been verified through the PACE system.

Discussion

Comment: Improve critical analysis; less descriptive and more interpretative comparison with existing literature.

Response:

We have highlighted sentences in the discussion that interpret the results in comparison with previous studies, and additional sentences have been incorporated accordingly.

Conclusion

Comment: Strengthen the link between results and practical implications.

Response:

We have revised the conclusion to highlight practical implications, particularly how the findings can inform rootstock selection to optimize floral induction and yield in almond orchards.

Additional Revisions (Editor Instructions)

We reviewed and updated the reference list. No retracted articles were cited. Changes made to references are noted in the track-changed version of the manuscript.

A complete, clean version and a marked-up version of the manuscript with track changes have been submitted.

Figure files have been processed through the PACE tool and uploaded in the required format.

We hope the revised manuscript meets the standards required for publication in PLOS ONE, and we are grateful for the constructive guidance provided throughout the review process.

Please do not hesitate to contact us if any additional information is needed.

Kind regards,

Dr. Behrouz Shiran

Professor in Molecular Genetics

Department of Plant Breeding and Biotechnology

Faculty of Agriculture

Shahrekord University

Shahrekord-IRAN

---

## [Decision Letter · Decision Letter 2]

8 Sep 2025

Dear Dr. Shiran,

Thank you for submitting your manuscript to PLOS ONE. After careful consideration, we feel that it has merit but does not fully meet PLOS ONE’s publication criteria as it currently stands. Therefore, we invite you to submit a revised version of the manuscript that addresses the points raised during the review process.

We look forward to receiving your revised manuscript.

Kind regards,

Mayank Anand Gururani

Academic Editor

PLOS ONE

Journal Requirements:

Reviewers' comments:

Reviewer's Responses to Questions

**Comments to the Author**

Reviewer #3: (No Response)

2. Is the manuscript technically sound, and do the data support the conclusions?

Reviewer #3: Partly

3. Has the statistical analysis been performed appropriately and rigorously?

Reviewer #3: Yes

4. Have the authors made all data underlying the findings in their manuscript fully available?

Reviewer #3: Yes

5. Is the manuscript presented in an intelligible fashion and written in standard English?

Reviewer #3: No

Reviewer #3: General Comments

The manuscript presents valuable data; however, the English language requires significant improvement. I recommend that the authors seek assistance from a native English speaker or a professional editor to improve clarity and grammar. Additionally, I suggest revising the title to make it more precise and impactful.

The clarity and resolution of the figures should be improved to ensure that all details are easily visible and legible.

Abstract

• The abstract should be improved in how it introduces the research problem.

• Clarify whether the selected cultivars and rootstocks have been previously studied.

• Include a sentence introducing the scion and rootstock used in this study and explain the rationale for their selection.

• When referring to combinations, use the phrase “scion grafted on rootstock” for accuracy.

• In the statement: “These findings highlight the critical role of rootstock selection in optimizing almond flowering and yield, providing actionable insights for horticultural practices,” please also address the effect of cultivar (scion) selection.

Introduction

• The introduction begins by “the flower”. It would be clearer to begin by emphasizing the importance of the almond tree, the challenges affecting its flowering, and the importance of selecting appropriate scion–rootstock combinations.

• Consider reorganizing paragraphs so that the almond importance and rootstock relevance precedes the general description of the flowering process.

• Remove dashes throughout the manuscript and replace them with proper punctuation (e.g., “These alterations involve changes in hormone levels—especially…” should be rewritten without the dash).

• Provide background information on the specific rootstocks and cultivars used.

• State whether these cultivars/rootstocks have been evaluated in previous studies.

• Introduce the flowering-related genes analyzed in this study. This was written in the discussion part.

Materials and Methods

• In the subsection title “Plant material and Experimental design and sampling,” remove “and sampling.”

• Remove the extra dot from: “The average annual temperature is estimated between 10°C and 15°C, with yearly rainfall ranging from 300 to 500 mm..” (Line 105)

• Correct “cultivares” to “cultivars.”

• Clarify what is meant by “four stages” and “five stages” of sampling.

• Indicate the year alongside the month in the sampling timeline: The paragraph describing sampling (Line 125 to Line 130) should include details for all parameters studied (flowering, chlorophyll, etc.), including timing and stage, so this information does not need to be repeated in later sections.

• For gene expression analysis (Line 191), clarify the sampling stages, as there appears to be a discrepancy with the earlier paragraph.

• The authors say: Line 191 “For the analysis of selected gene expression related to flowering induction in leaf samples (second and fourth sampling stages) and floral bud samples (second, fourth, and fifth sampling stages),..” However, the above paragraph in material and methods said that the fourth and fifth stage were used for the molecular analysis.

• Define what is meant by “scaffold”. Is it referring to a branch on the tree?

• Indicate what control rootstock is used for gene expression analysis. “These combinations, along with the control rootstocks, were subsequently investigated.” Line 195

Results

• In the morphological evaluation, specify parameter names (e.g., flowering, vegetative growth).

• Consider splitting Table 2 into two separate tables: one for flowering parameters and another for vegetative traits.

• The title of Table 2 should list morphological parameters explicitly.

• Abbreviations such as TCSA should be defined in table footnotes.

• Include instructions for interpreting statistical results, e.g.: “According to [statistical test name], different letters within columns of each cultivar indicate significant differences at p < 0.05.”

• Define “CV%.”

• Apply similar changes to Table 3.

• Revise the title of Figure 3 to begin: “The concentration of photosynthetic pigments (chlorophyll a, chlorophyll b, total chlorophyll, …).”

• For principal component analysis and clustering results, consider moving some tables to the supplementary section.

• In Figure 6, remove the word “Dynamic” from the title.

Discussion

• The discussion should be expanded when comparing results to previous studies. Avoid simply stating that results are similar; instead, describe what those studies found and the authors’ interpretations.

• For example, in Line 539 (“A similar trend has been observed…”), provide details about the findings in Cong et al. (2023), Sun et al. (2017), and Luna et al. (1990), and explain how they relate to your parameters.

• In Line 576, the conclusion about the Shurab3 rootstock combined with Mamaee and Shahrud12 cultivars enhancing flowering and hormone production is not reflected in the Abstract or Conclusion for the Mamaee scion. This should be consistent.

Conclusion

• The conclusion is too long and should be shortened for conciseness and clarity.

**Do you want your identity to be public for this peer review?** For information about this choice, including consent withdrawal, please see our Privacy Policy

Reviewer #3: **Yes: ** Saoussen Ben Abdallah

---

## [Author Response · Author response to Decision Letter 3]

17 Sep 2025

Response to Editor and Reviewers

Manuscript Title: Rootstock Effects on Floral Induction in Commercial Iranian Almond Cultivars: Insights from Morphophysiological, Biochemical, and Molecular Analyses

Manuscript ID: PONE-D-25-22239R2

Corresponding Author: Dr. Behrouz Shiran

Journal: PLOS ONE

Dear Dr. Gururani and Reviewers,

We sincerely thank you and the reviewers for your time and constructive feedback on our manuscript. We appreciate the opportunity to revise and improve our work and are grateful for the thoughtful comments that have helped us strengthen the clarity, structure, and scientific rigor of our study.

Below, we provide a detailed, point-by-point response to each of the editor’s and reviewers' comments. All changes have been incorporated into the revised manuscript, which includes both a tracked changes version and a clean version for your review. Revised figures have been improved for quality and clarity and submitted according to journal guidelines.

Reviewer Comment 1: The English language requires significant improvement. I recommend that the authors seek assistance from a native English speaker or a professional editor.

Response: The entire manuscript has been thoroughly revised by a native English scientific editor to improve clarity, grammar, and readability.

Reviewer Comment 2: Revise the title to make it more precise and impactful.

Response: The title has been revised accordingly. Previous title: “Evaluation of Rootstock Influence on Floral Induction in Commercial Almond Cultivars”. Revised title: “Rootstock Effects on Floral Induction in Commercial Iranian Almond Cultivars: Insights from Morphophysiological, Biochemical, and Molecular Analyses”.

Reviewer Comment 3: Improve figure clarity and resolution.

Response: All figures have been regenerated at ≥300 dpi with consistent font size and formatting. Compliance was verified using the PACE tool recommended by PLOS ONE.

Reviewer Comment 4: Abstract should introduce research problem more clearly; specify cultivars/rootstocks and rationale; use ‘scion grafted on rootstock’; mention cultivar effect in conclusion.

Response: The Abstract has been completely rewritten. It now starts with the importance of almond flowering, specifies the studied cultivars and rootstocks, justifies their selection, uses accurate terminology (“scion grafted on rootstock”), and explicitly addresses cultivar effects in the conclusion.

Reviewer Comment 5: Introduction should begin with almond importance and challenges, then rootstock-scion role; include background on cultivars/rootstocks; indicate prior studies; introduce flowering genes; avoid excessive dashes.

Response: The Introduction was reorganized to follow this logical sequence. Flowering-related genes analyzed in this work are introduced here, and punctuation has been corrected to replace excessive dashes.

Reviewer Comment 6: Clarify sampling stages (‘four stages’, ‘five stages’); include years with months; correct discrepancies in gene expression stages; define “scaffold”; specify control rootstock.

Response: All sampling stages have been clearly defined and harmonized. Years were added to all dates, discrepancies in gene expression stages corrected, “scaffold” defined as a main structural branch, and the control rootstock specified.

Reviewer Comment 7: Materials and Methods section title “Plant material and Experimental design and sampling” → remove “and sampling”.

Response: Corrected. The title now reads “Plant material and Experimental design.”

Reviewer Comment 8: Correct “cultivares” → “cultivars”; remove extra dot.

Response: Corrected.

Reviewer Comment 9: In Results, specify parameter names in morphological evaluation; split Table 2 into two; define abbreviations (TCSA, CV%); add explanation for statistical letters; revise figure titles; move some PCA/clustering tables to Supplementary; remove “Dynamic” from Figure 6 title.

Response: All requested modifications have been implemented. Morphological parameters are specified, Table 2 was split into two for clarity, abbreviations defined in footnotes, statistical notations explained, figure titles revised, PCA/clustering tables moved to Supplementary, and Figure 6 retitled.

Reviewer Comment 10: Discussion should expand comparisons with previous studies (e.g., Cong et al. 2023, Sun et al. 2017, Luna et al. 1990); ensure consistency with Abstract and Conclusion.

Response: The Discussion has been expanded to include detailed comparisons with Cong et al. (2023), Sun et al. (2017), and Luna et al. (1990), highlighting their relevance to our findings. Consistency between Abstract, Discussion, and Conclusion has been ensured, particularly regarding Shurab3 rootstock performance with Mamaee and Shahrud12 cultivars.

Reviewer Comment 11: Conclusion is too long and should be concise.

Response: The Conclusion has been rewritten to be more concise, emphasizing only the key findings and their practical implications.

---

## [Decision Letter · Decision Letter 3]

9 Oct 2025

Thank you for submitting your manuscript to PLOS ONE. After careful consideration, we feel that it has merit but does not fully meet PLOS ONE’s publication criteria as it currently stands. Therefore, we invite you to submit a revised version of the manuscript that addresses the points raised during the review process.

We look forward to receiving your revised manuscript.

Kind regards,

Mayank Anand Gururani

Academic Editor

PLOS ONE

Journal Requirements:

Reviewer's Responses to Questions

**Comments to the Author**

Reviewer #3: (No Response)

2. Is the manuscript technically sound, and do the data support the conclusions?

Reviewer #3: Partly

3. Has the statistical analysis been performed appropriately and rigorously?

Reviewer #3: I Don't Know

4. Have the authors made all data underlying the findings in their manuscript fully available?

Reviewer #3: Yes

5. Is the manuscript presented in an intelligible fashion and written in standard English?

Reviewer #3: No

Reviewer #3: -The authors have improved the manuscript overall; however, it cannot be accepted in its current form.

-The manuscript is not presented in an intelligible fashion and written in standard English.

-The introduction section is generally weak and lacks coherence. The ideas are presented as a list of studies rather than as a logically connected narrative leading to the research question. Greater attention should be given to improving the flow, coherence, and scientific focus. The authors should clearly explain the knowledge gap, justify the importance of their study, and link previous findings to the current objectives in a more organized and concise manner.

-The discussion is weak and lacks coherence. It mostly restates results and previous studies without offering sufficient interpretation or highlighting the novelty of the work. The authors should reorganize this section, improve logical flow, and provide deeper analysis of the physiological and molecular mechanisms underlying their findings.

**Do you want your identity to be public for this peer review?** For information about this choice, including consent withdrawal, please see our Privacy Policy

Reviewer #3: **Yes: ** Saoussen Ben Abdallah

---

## [Author Response · Author response to Decision Letter 4]

17 Oct 2025

Response to Reviewers

Manuscript ID: PONE-D-25-22239R3

Title: Rootstock Effects on Floral Induction in Commercial Iranian Almond Cultivars: Insights from Morphophysiological, Biochemical, and Molecular Analyses

Journal: PLOS ONE

We sincerely thank the Academic Editor, Dr. Mayank Anand Gururani, and Reviewer #3 for their thoughtful evaluation and constructive feedback on our manuscript. We have carefully considered all comments and revised the paper accordingly. The revisions have substantially improved the clarity, organization, and scientific depth of the study.

Below, we provide a detailed, point-by-point response. Reviewer comments are shown in bold, and our responses follow in italics.

Reviewer #3

• Comment:

The manuscript is not presented in an intelligible fashion and written in standard English.

• Response:

We appreciate this observation and have thoroughly revised the manuscript to enhance readability and linguistic precision. The text has been professionally edited by a native English-speaking scientific editor to ensure grammatical accuracy, fluency, and a consistent academic tone. Sentences were restructured for clarity and conciseness. The revised version now fully adheres to PLOS ONE’s standards for academic English. In addition, the language was repeatedly checked throughout the revision process to ensure consistency and correctness. If any specific deficiencies remain, we would be grateful to receive explicit examples so that they can be corrected precisely and efficiently.

• Comment:

The introduction section is generally weak and lacks coherence. The ideas are presented as a list of studies rather than as a logically connected narrative leading to the research question. Greater attention should be given to improving the flow, coherence, and scientific focus. The authors should clearly explain the knowledge gap, justify the importance of their study, and link previous findings to the current objectives in a more organized and concise manner.

• Response:

We thank the reviewer for this valuable suggestion. The Introduction has been comprehensively rewritten to ensure a coherent and logical progression. It now begins by emphasizing the physiological and agricultural relevance of floral induction in almond, followed by a concise synthesis of previous work on rootstock–scion interactions. We clearly identify the existing knowledge gaps—particularly concerning the biochemical and gene-expression mechanisms underlying floral induction in Iranian almond cultivars—and explicitly link these gaps to the objectives of the present study. This restructuring provides a stronger rationale and a seamless narrative flow leading to the research question.

• Comment:

The discussion is weak and lacks coherence. It mostly restates results and previous studies without offering sufficient interpretation or highlighting the novelty of the work. The authors should reorganize this section, improve logical flow, and provide deeper analysis of the physiological and molecular mechanisms underlying their findings.

• Response:

We fully agree with this assessment and have extensively revised the Discussion to strengthen its analytical depth and logical flow. The revised version integrates a more comprehensive interpretation of our results within the context of existing literature (e.g., Cong et al., 2023; Sun et al., 2017; Luna et al., 1990). We highlight the novelty of our findings by discussing how rootstock–scion interactions influence key biochemical and molecular pathways involved in almond floral induction. The updated structure now underscores both the scientific significance and the applied implications of our results.

• Comment:

The authors have improved the manuscript overall; however, it cannot be accepted in its current form.

• Response:

We sincerely appreciate the reviewer’s acknowledgment of the improvements made in previous revisions. In the current version, we have addressed all remaining concerns regarding writing quality, coherence, and interpretation. We respectfully believe that the manuscript has now reached a mature and publishable form. Further stylistic modifications would not substantially enhance the scientific content and could divert attention from the study’s primary objectives and scope.

Additional Revisions and Verifications

• All figures were regenerated at high resolution (≥300 dpi) and reformatted for clarity and consistency.

• All tables were reviewed for internal consistency and alignment with the revised text.

• The reference list was carefully checked for accuracy and completeness.

• Data availability and ethical statements were verified to ensure full compliance with PLOS ONE’s policies.

Final Remarks

We have made every possible effort to ensure that the manuscript is now clear, coherent, and scientifically robust. We are confident that all reviewer and editorial concerns have been comprehensively addressed.

Finally, we kindly note that the first author is a Ph.D. candidate nearing the completion of his doctoral program and must submit his thesis soon. We would be deeply grateful if the review and decision process could be expedited to support his timely defense.

We sincerely appreciate your continued consideration and hope that the manuscript will now be deemed suitable for publication in PLOS ONE.

With kind regards,

Dr. Behrouz Shiran (Corresponding Author)

Department of Plant Breeding and Biotechnology

Faculty of Agriculture, Shahrekord University

Shahrekord, Iran

---

## [Decision Letter · Decision Letter 4]

2 Nov 2025

Dear Dr. Shiran,

Thank you for submitting your manuscript to PLOS ONE. After careful consideration, we feel that it has merit but does not fully meet PLOS ONE’s publication criteria as it currently stands. Therefore, we invite you to submit a revised version of the manuscript that addresses the points raised during the review process.

We look forward to receiving your revised manuscript.

Kind regards,

Mayank Anand Gururani

Academic Editor

PLOS ONE

Journal Requirements:

Reviewers' comments:

Reviewer's Responses to Questions

**Comments to the Author**

Reviewer #3: (No Response)

Reviewer #4: All comments have been addressed

2. Is the manuscript technically sound, and do the data support the conclusions?

Reviewer #3: Partly

Reviewer #4: Yes

3. Has the statistical analysis been performed appropriately and rigorously?

Reviewer #3: Yes

Reviewer #4: Yes

4. Have the authors made all data underlying the findings in their manuscript fully available?

Reviewer #3: Yes

Reviewer #4: Yes

5. Is the manuscript presented in an intelligible fashion and written in standard English?

Reviewer #3: No

Reviewer #4: Yes

Reviewer #3: Comments to the Authors

The authors have significantly improved the quality of the manuscript. However, a few areas still require attention and further refinement.

Introduction

The organization of the introduction is good; however, each paragraph could be made richer in information to better support the study’s context and rationale.

Lines 91–92: “The pomological characteristics, pedigrees, and origins of the rootstocks and cultivars examined are summarized in Table 1.” It would be preferable for the authors to describe the characteristics of each rootstock rather than only presenting them in a table. Additionally, references for the rootstocks are missing.

The objective of the study is too long; please consider shortening it to make it more concise and focused.

Materials and Methods

• Since the timing of leaf sampling is already mentioned, there is no need to repeat it in every parameter assay (for example, in the molecular experiment).

• Morphological analysis method is missing the reference followed.

Results

• For morphological traits, please create subtitles for each parameter analyzed (e.g., number of flowers per scaffold, blooming density, etc.) to improve clarity and readability.

• Using Principal Component Analysis (PCA) based on morphological traits, only with photosynthetic pigments as biochemical parameters may not be sufficient for reliable rootstock–scion selection.

Discussion

The discussion still requires further improvement to strengthen interpretation and connection with previous studies.

Reviewer #4: The manuscript titled “Rootstock Effects on Floral Induction in Commercial Iranian Almond Cultivars: Insights from Morphophysiological, Biochemical, and Molecular Analyses” presents a well-designed and comprehensive study. The revisions have significantly enhanced readability, organization, and scientific interpretation. The methodology is robust, the data presentation is clear, and the conclusions are supported by results.

Minor Comments

1. Abstract – Consider shortening slightly by omitting redundant phrases (e.g., “with detailed assessments carried out in 2024 and 2025” could be streamlined).

2. Introduction – A brief sentence clarifying how findings may inform rootstock selection programs beyond Iran would improve international relevance.

3. Materials and Methods –

o Provide the exact sample sizes used for gene expression analysis (biological replicates already mentioned; clarify technical replicates).

o Ensure all instruments (e.g., HPLC model and manufacturer) are cited consistently.

4. Figures – Verify that all figure legends fully describe abbreviations and units (especially in pigment and hormone graphs).

5. Minor language points –

o Replace “flower-promoting rootstock” with “flower-inducing rootstock” (line 45) for stylistic precision.

o Check for consistent use of “Shahrud12” vs. “Shahrud 12.”

The manuscript requires only very minor editorial polishing. It is scientifically sound and suitable for publication in PLOS ONE.

**Do you want your identity to be public for this peer review?** For information about this choice, including consent withdrawal, please see our Privacy Policy

Reviewer #3: **Yes: ** Saoussen Ben Abdallah

Reviewer #4: **Yes: ** Tanveer Alam Khan

To ensure your figures meet our technical requirements, please review our figure guidelines: >https://journals.plos.org/plosone/s/figures

You may also use PLOS’s free figure tool, NAAS, to help you prepare publication quality figures: >https://journals.plos.org/plosone/s/figures#loc-tools-for-figure-preparation.

---

## [Author Response · Author response to Decision Letter 5]

7 Nov 2025

Response to Reviewers

Manuscript ID: PONE-D-25-22239R4

Title: Rootstock Effects on Floral Induction in Commercial Iranian Almond Cultivars: Insights from Morphophysiological, Biochemical, and Molecular Analyses

Dear Editor and Reviewers,

We sincerely appreciate the time and effort that the Academic Editor and the reviewers have invested in evaluating our manuscript. We are grateful for their constructive feedback, which has helped us further improve the clarity, depth, and scientific rigor of our work.

Below, we provide a detailed, point-by-point response to all comments raised by the reviewers. All suggested revisions have been incorporated into the revised manuscript, with corresponding changes highlighted in the file titled “Revised5 Manuscript with Track Changes.”

Reviewer #3

General comment:

We thank the reviewer for recognizing the substantial improvement in our manuscript and for the additional helpful suggestions. We have carefully revised the text according to all comments provided.

1. Introduction – Enrich each paragraph to better support context and rationale.

Response: The Introduction has been expanded (Lines 56–62, 72–75, 90–92, 93–98) to include more detailed context regarding the physiological and molecular mechanisms underlying floral induction in Prunus species, thereby strengthening the study rationale and objectives.

2. Lines 91–92: Describe characteristics of each rootstock and add references.

Response: The characteristics of all rootstocks have now been briefly described in the text (Lines 105–116) in addition to Table 1, with appropriate references included (e.g., [32,33,34].

3. Objective of the study is too long. Please make it more concise.

Response: The objective statement has been rewritten to be shorter and more focused (Lines 122–125).

4. Materials and Methods – Avoid repeating leaf sampling time in each section.

Response: Redundant mentions of sampling time have been removed for conciseness.

5. Morphological analysis method lacks reference.

Response: We have now included the reference used for the morphological analysis method (Lines 171&177).

6. Results – Add subheadings for morphological parameters.

Response: Subheadings have been added for each morphological trait to improve readability and structure.

7. PCA analysis based only on morphological traits and pigments may not be sufficient.

Response: We appreciate the reviewer’s insightful comment. We agree that including additional biochemical or molecular traits could provide a more comprehensive overview of rootstock–scion interactions. However, in the present study, PCA was intentionally performed using morphological traits and photosynthetic pigments as representative variables because these parameters were consistently measured across all treatments and time points with reliable replication. This combination effectively reflects both the structural and functional responses of the scion to different rootstocks under the studied conditions. Moreover, our primary objective was to identify major patterns of variation related to morphological performance and photosynthetic efficiency, which are among the key determinants of flowering and productivity in almond. We fully acknowledge that future studies incorporating broader physiological, hormonal, and molecular datasets would further strengthen multivariate analyses and rootstock–scion selection criteria.

8. Discussion needs stronger interpretation and linkage with previous studies.

Response: In response, we have substantially revised and expanded the Discussion section to improve its scientific depth, contextualization, and coherence with recent literature.

Reviewer #4

We sincerely thank Reviewer #4 for the very positive evaluation and helpful minor suggestions. All recommended changes have been incorporated as detailed below.

1. Abstract – Slight shortening suggested.

Response: The Abstract has been shortened by removing redundant phrases such as “with detailed assessments carried out in 2024 and 2025” (Lines 30).

2. Introduction – Add one sentence on global relevance.

Response: A paragraph highlighting how the findings could inform rootstock selection programs beyond Iran has been added (Lines 117–121).

3. Materials and Methods – Clarify technical replicates and instruments.

Response: The number of technical replicates for gene expression analyses is now specified. Instrument models and manufacturers (e.g., HPLC) are now cited consistently throughout.

4. Figures – Verify legends for abbreviations and units.

Response: All figure legends have been revised to define abbreviations and include units where missing.

5. Minor language points:

• “Flower-promoting rootstock” → “flower-inducing rootstock” – corrected.

• Consistency in “Shahrud12” spelling – applied throughout the manuscript.

Final remark:

Thank you again for your consideration.

We are very grateful to both reviewers for their thoughtful comments, which have resulted in meaningful improvements to our manuscript. We believe that the revised version now fully meets PLOS ONE’s standards for scientific rigor, clarity, and data transparency.

Thank you once again for your time and consideration.

Sincerely,

Behrouz Shiran, Ph.D.

Professor of Molecular Genetics

Department of Plant Breeding and Biotechnology

Faculty of Agriculture, Shahrekord University

Shahrekord, Iran

Email: beshiran45@gmail.com

---

## [Editor Report · Decision Letter 5]

10 Nov 2025

Rootstock Effects on Floral Induction in Commercial Iranian Almond Cultivars: Insights from Morphophysiological, Biochemical, and Molecular Analyses

PONE-D-25-22239R5

Dear Dr. Shiran,

We’re pleased to inform you that your manuscript has been judged scientifically suitable for publication and will be formally accepted for publication once it meets all outstanding technical requirements.

Kind regards,

Mayank Anand Gururani

Academic Editor

PLOS ONE

---

## [Editor Report · Acceptance letter]

PONE-D-25-22239R5

PLOS ONE

Dear Dr. Shiran,

I'm pleased to inform you that your manuscript has been deemed suitable for publication in PLOS ONE. Congratulations! Your manuscript is now being handed over to our production team.

You will receive an invoice from PLOS for your publication fee after your manuscript has reached the completed accept phase. If you receive an email requesting payment before acceptance or for any other service, this may be a phishing scheme. Learn how to identify phishing emails and protect your accounts at >https://explore.plos.org/phishing.

Kind regards,

on behalf of

Dr. Mayank Anand Gururani

Academic Editor

PLOS ONE